



# Potential based Thermodynamics with Consistent Conservative Cascade Transport for Implicit Large Eddy Simulation: PTerodaC$^3$TILES version 1.0

John Thuburn[1]

[1]Department of Mathematics and Statistics, University of Exeter, Exeter, EX4 4QF, UK

**Correspondence:** John Thuburn (j.thuburn@exeter.ac.uk)

**Abstract.** A new computational fluid dynamics code for Large-Eddy Simulation (LES) of the atmospheric boundary layer and convection is presented and made available. A key novelty is that moist thermodynamics is formulated in terms of thermodynamic potentials, ensuring thermodynamic consistency. Despite the apparent complexity of the thermodynamic potential approach, the model's performance demonstrates that it is feasible and effective at reasonable computational cost for three-dimensional simulations. Semi-implicit semi-Lagrangian numerical methods are used; such methods are unusual for simulating boundary layer and convective flows and are more typical of global atmospheric models. Moreover, the model includes no explicit scheme to represent subgrid-scale fluxes of scalars and momentum but relies instead on the mixing and dissipation resulting from the numerical methods used; in other words, it employs Implicit LES (ILES). Sample results from several standard LES test cases show that the model's ability to capture the main aspects of the flows is comparable to other LES models. At the same time, the results highlight limitations of the ILES approach near the bottom boundary and suggest that ILES might need to be augmented in some way, for example, by distributing the convergence of surface fluxes over several model layers. Also, results for a marine stratocumulus case show a significant sensitivity to different options for the numerical methods and parameters used. Further development and application of the code would benefit from a deeper understanding of both the bottom boundary behaviour and the sensitivities to numerics.

## 1 Introduction

Large-Eddy Simulation (LES) has been an invaluable computational tool in atmospheric science since the early 1970's, both for advancing our understanding of complex atmospheric processes such as boundary-layer turbulence and convection and for informing the development of parameterizations of those processes for use in weather and climate models. This article presents the formulation of a new LES code developed by the author with a three-fold motivation:

1. to demonstrate the feasibility of using thermodynamic potentials to achieve a consistent representation of moist thermodynamics in a three-dimensional fluid dynamics code;





2. to evaluate the use in LES of the sort of numerical methods more usually used in synoptic and global-scale models;

3. to provide a modelling tool that is easy to set up and run and in which it is easy to set up new test cases and diagnostics.

The thermodynamics of moist air is complicated, and atmospheric models often make approximations. Some common approximations introduce inconsistencies between different aspects of the system or with the laws of thermodynamics—see Thuburn (2017b) for examples. Thermodynamic consistency can be ensured by deriving all thermodynamic quantities from one of the four standard thermodynamic potentials, internal energy, enthalpy, Helmholtz free energy, or Gibbs function, expressed as a function of its natural variables. Provided any approximations are made directly to the thermodynamic potential before
deriving other quantities, consistency is maintained. This approach has been proposed for use in ocean modelling for consistent treatment of the complex equation of state of seawater (IOC et al., 2010). More recently, the approach has been advocated for deriving consistent equation sets for the thermodynamics of moist air in atmospheric models (Vallis, 2017; Eldred et al., 2022; Staniforth, 2022). As an alternative, a model can be formulated and implemented directly in terms of the thermodynamic potential and its derivatives (Thuburn, 2017b; Bowen and Thuburn, 2022a, b); in this way different approximations to the
thermodynamics can be implemented by modifying the minimal number of routines.

For the case of thermodynamic equilibrium (no supersaturation, no condensate in subsaturated air, all components and phases at the same temperature) and in the absence of ice, Thuburn (2017b) presented a semi-implicit semi-Lagrangian model solving the compressible Euler equations in which the moist thermodynamics was formulated in terms of the Gibbs function for moist air. Whilst encouraging, this initial implementation suffered several limitations. First, because the formulation works in
terms of the total Gibbs function for moist air, the water content must be partitioned into vapour and condensate whenever the Gibbs function is evaluated, complicating the calculation. More importantly, since the natural variables for the Gibbs function pressure $p$ and temperature $T$ are intensive variables, knowledge of $p$ and $T$ (and total specific humidity $q$) alone is insufficient to completely determine the equilibrium state, particularly the partition of water into its three phases, at the triple point. Thus, the implementation is restricted to two phases: vapour and liquid. Finally, the formulation of Thuburn (2017b) could not
represent important nonequilibrium effects such as the delayed freezing of supercooled cloud droplets or the evaporation of rain in subsaturated air.

These limitations were overcome by Bowen and Thuburn (2022a, b). They formulated the thermodynamics in terms of the internal energy, whose natural variables are the extensive variables specific volume $\alpha$ and specific entropy $\eta$, avoiding the difficulty at the triple point. Moreover, by working with the individual internal energy potentials for dry air, water vapour, liquid
water, and ice, rather than a combined thermodynamic potential for the air parcel, they were able to separate the calculation of the potentials from the calculation of the air parcel equilibrium state, simplifying the formulation. By expressing the evolution of a subset of variables in terms of thermodynamic forces and a resistivity matrix, they were able to account for departures from thermodynamic equilibrium, while seamlessly approaching the equilibrium case in the limit of zero resistivity.

The codes developed by Thuburn (2017b) and Bowen and Thuburn (2022a, b) were two-dimensional vertical slice models
and they were applied to simple buoyant bubble test problems. The apparent complexity of the thermodynamic potential approach, with accompanying concerns about its computational cost, might discourage model developers from pursuing the





approach in three-dimensional models. A primary goal of the work described here is is to demonstrate that the approach can be applied successfully, without excessive expense, in a three-dimensional model suitable for studying complex boundary-layer and convective flows.

Traditionally, LES models for atmospheric applications are often based on relatively simple numerical methods supplemented by more or less sophisticated subgrid models (e.g. Siebesma et al., 2003). Typically, advection schemes are Eulerian and time stepping is explicit, so that stability requires the advective and diffusive Courant numbers to be less than some threshold value of order 1. Global weather and climate models, on the other hand, often use sophisticated (and relatively expensive) advection schemes that are stable for large advective Courant numbers (e.g. Temperton et al., 2001; Lin, 2004; Wood et al.,

2014; Melvin et al., 2024), though they require the deformational Courant number to be bounded. Bartello and Thomas (1996) have argued that such large-timestep advection schemes are no longer cost-effective in flow regimes where the energy spectrum is shallower, the Lagrangian and Eulerian timescales become more comparable, and the deformational Courant number is much closer to the advective Courant number. Nevertheless, traditional LES codes are often run with time steps more than an order of magnitude smaller than could be used by a semi-implicit semi-Lagrangian scheme at the same resolution (e.g. Stevens et al.,

2005, and compare section 5 below). This observation, combined with recent progress in improving the efficiency of conservative semi-implicit semi-Lagrangian solvers (Thuburn, 2024), encouraged the author to revisit the question by implementing a semi-implicit semi-Lagrangian LES model.

    On a closely related point, global models with sophisticated advection schemes sometimes do not include a subgrid model to handle the turbulent downscale cascades of potential enstrophy and energy but rely instead on the dissipative nature of the

75 advection scheme to play that role (e.g. Walters et al., 2017; ECMWF, 2023). In other words, they use a form of Implicit Large-Eddy Simulation or ILES (e.g. Margolin et al., 2006; Grinstein et al., 2007). Although there have been some pioneering attempts to use or evaluate ILES for boundary-layer and convective scale atmospheric flows (Margolin et al., 1999; Brown et al., 2000; Smolarkiewicz and Prusa, 2002), and there is growing interest (e.g. Pressel et al., 2017; Souza et al., 2023), the approach is still far from mainstream. However, the need to understand the strengths and weaknesses of ILES for convective

scale flows is becoming increasingly pressing as global prediction models begin to be used at kilometre-scale resolution (Satoh et al., 2008; Stevens et al., 2019; Hohenegger et al., 2023; Tomassini et al., 2023). PTerodaC$^3$TILES includes no subgrid model and uses the ILES approach.

    Finally, the author perceived a need for an LES tool that could be set up to run 'production' science with minimal effort on a desktop machine or even a laptop. PTerodaC$^3$TILES v1.0 comprises a single stand-alone fortran code and a namelist file.

No external packages are needed other than a fortran compiler (with OpenMP shared memory parallel capability if desired). Initial data and forcings for a number of standard cases are available just by selecting namelist options; new cases can easily be implemented by using the routines for existing cases as templates. To avoid large volumes of output, diagnostics are calculated 'online' at run time. Many standard diagnostics are available via namelist switches; others can easily be implemented and output using existing routines as templates. Further details are given in the User Manual, available from Zenodo; see the

*Code availability* section.





Some aspects of the model formulation are novel or noteworthy; these are introduced briefly here and discussed in more detail in section 2.

Unlike most atmospheric models, changes of phase of water and latent heat release are not treated as separate 'physics' source terms, but are fully integrated within the dynamical core's semi-implicit semi-Lagrangian time stepping (Thuburn, 2017b; Thuburn et al., 2022; Bowen and Thuburn, 2022a, b, sections 2.1, 2.3 below). To maintain consistency of the thermodynamics, surface fluxes of water imply surface fluxes of mass. The same is true for the somewhat artificial sources of water specified in the domain interior for some test cases. Because of the model's Charney-Phillips vertical staggering, with total density stored at $p$-levels and specific humidities stored at $w$-levels, special care is needed to maintain that consistency (section 2.7). Mass sources accompanying water sources are neglected in most atmospheric models.

In contrast to Thuburn (2017b) and Bowen and Thuburn (2022a, b), in PTerodaC$^3$TILES conservative options are available for the advection of moisture and entropy variables. Although the numerical methods do not exactly conserve energy, energy conservation is significantly improved as a side-effect of a conservative treatment of entropy and water (Thuburn, 2022).

Even with the use of cheap advection updates during the main solver iterations (Thuburn, 2024), advection remains one of the most expensive components of the model (along with the elliptic solver). Some modifications are made to the SLICE conservative semi-Lagrangian advection scheme (Zerroukat et al., 2009) to improve its efficiency, particularly to minimize the use of conditional code by avoiding 'searching'. Geometrical calculations of coordinate line intersections to determine 'intermediate departure points' are replaced by additional trajectory calculations, and information generated in remapping volume and mass is re-used in remapping other fields (section 2.6).

Linearization of the thermodynamics leads to an $11 \times 14$ linear subsystem at each model gridpoint that must be diagonalized in order to build the Helmholtz problem and to enable backsubstitution for the semi-implicit time stepping. To reduce what would otherwise potentially be a significant computational expense, the equations and unknowns are re-ordered to exploit the moderate sparsity of the thermodynamic subsystem, which is essentially the same at all gridpoints. The cost of the diagonalization is thereby reduced to about 30% of the cost of a full Gaussian elimination for the case of equilibrium thermodynamics. The cost is further reduced by carrying out the diagonalization on all matrices in a grid column at the same time, with the inner loop over vertical levels, to improve vectorization (section 2.10).

Some of the test cases implemented specify the use of Monin-Obukhov theory to compute surface momentum fluxes. Appendix A4 presents a slight reformulation of Monin-Obukhov theory that, with the aid of some curve-fitting, enables the friction velocity $U_*$ to be obtained without the need for an iterative calculation and guarantees the existence of a unique solution for $U_*$ even when the validity of Monin-Obukhov theory breaks down.

Section 4 summarizes some of the ways in which the correctness of the formulation and implementation have been verified. Section 5 presents some sample results from standard LES test cases to demonstrate the performance of the model. The conclusions and areas where further work is needed are discussed in section 6.





## 2 Model formulation

The formulation of PTerodaC$^3$TILES is inspired by semi-implicit semi-Lagrangian dynamical cores such as ENDGame (Wood
et al., 2014) and GungHo (Melvin et al., 2024) that iterate towards a (possibly off-centred) Crank-Nicolson time discretization
along trajectories. Such a formulation has been found to be stable and robust at large time steps for synoptic-scale flow. A
significant departure, though, is in the treatment of moist thermodynamics. First, to guarantee consistency, the thermodynamics
is expressed in terms of thermodynamic potentials, in this case the internal energy. Second, processes such as phase changes and
latent heat release are not treated as separate 'physics' source terms but directly couple to the dynamics through the dependence
of pressure and buoyancy on the thermodynamic state. The wide range of timescales associated with the thermodynamics,
ranging from instantaneous for processes in equilibrium to many minutes for some nonequilibrium processes, is naturally
handled by the semi-implicit time discretization.

A feature of the consistent treatment of thermodynamics is that any surface or interior source of moisture implies a corresponding source of mass (section 2.7). Such a mass source is neglected in most atmospheric models.

PTerodaC$^3$TILES version 1.0 includes only equilibrium thermodynamics, but almost all of the machinery is in place to
deal with the nonequilibrium case. Also, this version includes no representation of microphysical processes; any condensate is
simply carried along with the flow. The inclusion of a simple microphysics scheme is a priority for future work.

### 2.1 Continuous governing equations

The continuous equations to be solved are the same as those solved by Bowen and Thuburn (2022a), extended to three spatial
dimensions, and with the inclusion of Coriolis terms and external source terms for mass, momentum and water. See that article
for a derivation and in-depth discussion. For the purpose of exposition it is convenient to split the governing equations into
dynamics and thermodynamics

### 2.1.1 Dynamics

The subset of governing equations involving material derivatives is the following:

$$\frac{1}{\mathcal{V}} \frac{\mathrm{D}}{\mathrm{D}t}(\mathcal{V}\rho) = S_\rho, \tag{1}$$

$$\frac{\mathrm{D}\boldsymbol{u}}{\mathrm{D}t} + 2\boldsymbol{\Omega} \times \boldsymbol{u} + \alpha \nabla p + \nabla \Phi = \boldsymbol{S_u}, \tag{2}$$

$$\frac{1}{\mathcal{V}} \frac{\mathrm{D}}{\mathrm{D}t}(\mathcal{V}\rho q) = \rho S_q, \tag{3}$$

$$\frac{1}{\mathcal{V}} \frac{\mathrm{D}}{\mathrm{D}t}(\mathcal{V}\rho \eta) = \rho \boldsymbol{J}^{\mathrm{T}} \boldsymbol{P} + \rho S_\eta, \tag{4}$$





$$\frac{\mathrm{D}\boldsymbol{X}}{\mathrm{D}t} = \boldsymbol{J} + \boldsymbol{S_X}. \tag{5}$$

Here, $\rho$ is the total fluid density, $\boldsymbol{u} = (u, v, w)$ is the fluid (barycentric) velocity, $q$ is the total specific humidity, $\eta$ is the total specific entropy, $\alpha = 1/\rho$ is the total specific volume, $p$ is the pressure, and $\boldsymbol{\Omega}$ is the rotation vector of the frame of reference. $\Phi = \mathrm{g}z$ is the geopotential with g the gravitational acceleration. $S$ and $\boldsymbol{S}$ terms indicate external sources.

The vector $\boldsymbol{X} = (q^{\mathrm{l}}, q^{\mathrm{f}}, q^{\mathrm{v}}\eta^{\mathrm{v}}, q^{\mathrm{l}}\eta^{\mathrm{l}}, q^{\mathrm{f}}\eta^{\mathrm{f}})^{\mathrm{T}}$ (superscript T meaning transpose) encodes the additional thermodynamic information that needs to be predicted in the nonequilibrium case. Superscripts d, v, l, and f indicate dry air, water vapour, liquid water, and frozen water, respectively. $\boldsymbol{J}$ is a vector of thermodynamic 'fluxes', with $\boldsymbol{P}$ the corresponding thermodynamic 'forces', defined below. The expression $\boldsymbol{J}^{\mathrm{T}}\boldsymbol{P}$ in (4) gives the entropy source per unit mass due to nonequilbrium processes[1].

$\mathrm{D}/\mathrm{D}t$ is the material derivative. In equations (1), (3) and (4), $\mathcal{V}$ is the material volume element. These equations are written in a form that lends itself to numerical solution using a conservative semi-Lagrangian scheme. The material derivative in (5) is written in a form that anticipates discretization using an interpolating semi-Lagrangian scheme, on the assumption that it will be sufficient to advect the total specific humidity and entropy conservatively, with a cheaper non-conserving scheme for the components of $\boldsymbol{X}$. Nevertheless, an option for conservative advection is available. In the equilibrium case, however, $\boldsymbol{J}$ is not needed (see (13) below) so advection of $\boldsymbol{X}$ becomes superfluous and is automatically switched off.

The user can also choose to include additional tracers stored either at $p$-levels or at $w$-levels (see section 2.2). Since these do not feed back on the dynamics or thermodynamics we largely omit them from further discussion.

### 2.1.2 Thermodynamics

The diagnostic equations describing the thermodynamics are as follows:

$$q^{\mathrm{v}}\alpha^{\mathrm{v}} - q^{\mathrm{d}}\alpha^{\mathrm{d}} = 0, \tag{6}$$

$$(1 - \delta^{\mathrm{l}})\lambda^{\mathrm{l}} - \delta^{\mathrm{l}}q^{\mathrm{l}} = 0, \tag{7}$$

$$(1 - \delta^{\mathrm{f}})\lambda^{\mathrm{f}} - \delta^{\mathrm{f}}q^{\mathrm{f}} = 0, \tag{8}$$

$$q^{\mathrm{v}} + q^{\mathrm{l}} + q^{\mathrm{f}} - q = 0, \tag{9}$$

$$q^{\mathrm{v}}\alpha^{\mathrm{v}} + q^{\mathrm{l}}\alpha^{\mathrm{l}} + q^{\mathrm{f}}\alpha^{\mathrm{f}} - \alpha = 0, \tag{10}$$

---

[1]This entropy source term does not include the effects of viscosity and mixing of air parcels; see section 4.6.





$$q^{\mathrm{d}}\eta^{\mathrm{d}} + q^{\mathrm{v}}\eta^{\mathrm{v}} + q^{\mathrm{l}}\eta^{\mathrm{l}} + q^{\mathrm{f}}\eta^{\mathrm{f}} - \eta = 0, \tag{11}$$

$$p + e_{\alpha}^{\mathrm{d}} + e_{\alpha}^{\mathrm{v}} = 0, \tag{12}$$

$$\mathbf{R}\boldsymbol{J} - (\boldsymbol{P} - \boldsymbol{C}) \equiv -\boldsymbol{R}_{\mathrm{PE}} = 0. \tag{13}$$

Where needed, the mass fraction of dry air is given by $q^{\mathrm{d}} = 1 - q$.

Equation (6) states that the water vapour and dry air occupy the same volume within an air parcel. Equations (9), (10), and (11) state that the total specific humidity equals $q$, the total specific volume equals $\alpha$, and the total specific entropy equals $\eta$.

Equation (12) expresses Dalton's law of partial pressures, and is key for coupling the thermodynamics to the dynamics. Here, $e^{\mathrm{d}}(\alpha^{\mathrm{d}}, \eta^{\mathrm{d}})$, $e^{\mathrm{v}}(\alpha^{\mathrm{v}}, \eta^{\mathrm{v}})$, $e^{\mathrm{l}}(\eta^{\mathrm{l}})$, and $e^{\mathrm{f}}(\eta^{\mathrm{f}})$ are the internal energy potentials for dry air, water vapour, liquid water, and frozen water, respectively, expressed as functions of their natural variables (Appendix A2). As in Bowen and Thuburn (2022a, b), condensate is assumed to be incompressible, with $\alpha^{\mathrm{l}}$ and $\alpha^{\mathrm{f}}$ specified constants, so we suppress the dependence of $e^{\mathrm{l}}$ on $\alpha^{\mathrm{l}}$ and of $e^{\mathrm{f}}$ on $\alpha^{\mathrm{f}}$, and the pressure within any condensate (for example, as needed to compute the Gibbs function) is equal to the total pressure of the surrounding gas. Subscripts $\alpha$ and $\eta$ on the species internal energies indicate partial derivatives with respect to the respective natural variables.

Equation (13) is a set of 'phenomenological equations' relating thermodynamic fluxes $\boldsymbol{J}$ to the thermodynamic forces $\boldsymbol{P}$ that push the system towards equilibrium (de Groot and Mazur, 1984). $\boldsymbol{R}_{\mathrm{PE}}$ is a residual that should be driven towards zero by the iterative solver. In the present system $\boldsymbol{P}$ is given by

$$\boldsymbol{P} = \nabla_{\boldsymbol{X}}\eta = \frac{1}{T^{\mathrm{d}}}\left(g^{\mathrm{v}} - g^{\mathrm{l}}, g^{\mathrm{v}} - g^{\mathrm{f}}, T^{\mathrm{d}} - T^{\mathrm{v}}, T^{\mathrm{d}} - T^{\mathrm{l}}, T^{\mathrm{d}} - T^{\mathrm{f}}\right)^{\mathrm{T}}, \tag{14}$$

where the gradient $\nabla_{\boldsymbol{X}}\eta$ is taken at constant mass and energy, $T^{i}$ is the temperature of species $i$, and $g^{i}$ is the Gibbs function for species $i$, $i \in \{\mathrm{d,v,l,f}\}$ (Appendix A2). $\mathbf{R}$ is a symmetric and positive semi-definite resistivity matrix controlling the rate at which an air parcel approaches thermodynamic equilibrium. $\boldsymbol{C} = \eta_{\mathrm{ref}}(\delta^{\mathrm{l}}\lambda^{\mathrm{l}}, \delta^{\mathrm{f}}\lambda^{\mathrm{f}}, 0, 0, 0)^{\mathrm{T}}$ is a vector of switched Lagrange multipliers, with $\delta^{\mathrm{l}}, \delta^{\mathrm{f}} \in \{0, 1\}$, used in enforcing the constraints that $q^{\mathrm{l}}$ and $q^{\mathrm{f}}$ must be non-negative. For example, if the thermodynamic forces imply evaporation of liquid but $q^{\mathrm{l}}$ is already zero, then the liquid constraint is switched on ($\delta^{\mathrm{l}} = 1$) and the solution for $\lambda^{\mathrm{l}}$ balances the relevant component of $\boldsymbol{P}$. Finally, equations (7) and (8) are a convenient way to express the 'complementarity conditions' (e.g. Nocedal and Wright, 2006) that either $q^{\mathrm{l}}$ or $\lambda^{\mathrm{l}}$ must be zero, and either $q^{\mathrm{f}}$ or $\lambda^{\mathrm{f}}$ must be zero. $\eta_{\mathrm{ref}}$ is an arbitrary reference value, here equal to $1000\mathrm{Jkg}^{-1}\mathrm{K}^{-1}$, introduced to ensure that equations (7) and (8) are dimensionally correct.

Bowen and Thuburn (2022b) show how $\mathbf{R}$ can be related to the thermal conductivity of air and the molecular diffusivity of water vapour in air for cloud droplets of a given radius. In PTerodaC$^3$TILES v1.0 the resistivity $\mathbf{R}$ is set to zero, imposing local thermodynamic equilibrium. Nevertheless, almost all of the machinery is in place to handle the nonequilibrium case.





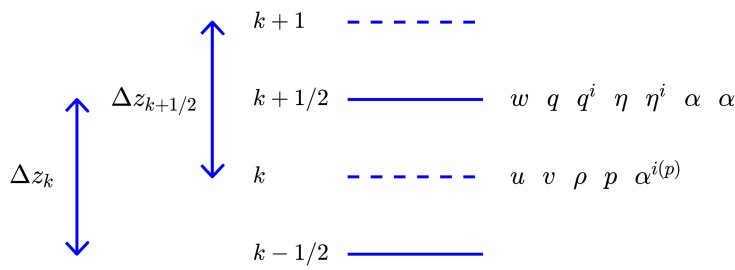

**Figure 1.** Schematic showing the placement of model variables on the vertically staggered Charney-Phillips grid. Superscript $i$ can stand for any of d, v, l, f.

Note that PTerodaC$^3$TILES has no explicit representation of subgrid variability in temperature or humidity, hence in condensate. Each grid cell is either entirely saturated or entirely unsaturated. In other words, it is an 'all-or-nothing' representation of saturation and condensate.

## 2.2 Domain, grid, and discretization

The domain of PTerodaC$^3$TILES v1.0 is rectangular and doubly periodic in the horizontal, with flat rigid boundaries at the bottom and top. Cartesian coordinates $(x, y, z)$ are aligned with the domain. When a full three-dimensional Coriolis force is used then the $x$- and $y$-directions are assumed to be East and North, respectively, but otherwise the model is agnostic about which direction is North.

A C-grid staggering (Arakawa and Lamb, 1977) is used in the horizontal and a Charney-Phillips grid staggering (Charney and Phillips, 1953) in the vertical (Fig. 1). The choice of a Charney-Phillips vertical grid is unusual for LES models but is more common in global models. It avoids computational modes—oscillatory vertical profiles of thermodynamic variables that spuriously satisfy discrete hydrostatic balance—and gives more accurate coupling between vertical velocity and buoyancy on small vertical scales, but makes it difficult to formulate an exactly energy-conserving scheme.

In the following, integer indices are used for $p$-points, with an offset of $1/2$ in the relevant direction for velocity points. Vertical indices 1 and $N_z$ correspond to the lowermost and uppermost $p$-levels; vertical indices $1/2$ and $N_z + 1/2$ correspond to the $w$-levels at the bottom and top model boundaries. Horizontal grid spacings $\Delta x$ and $\Delta y$ are uniform but need not be equal to each other. The vertical grid spacing $\Delta z$ may be uniform or stretched. See Appendix A1 for the grid specification in the stretched case. All of the results shown in sections 4 and 5 use uniform $\Delta z$.

Simple finite difference or finite volume approximations are used for gradient and divergence operators. The components of the pressure gradient (and similarly the geopotential gradient) are needed at velocity points:

$$\left.\frac{\partial p}{\partial x}\right|_{i+1/2\,j\,k} = \frac{p_{i+1\,j\,k} - p_{i\,j\,k}}{\Delta x}, \qquad \left.\frac{\partial p}{\partial y}\right|_{i\,j+1/2\,k} = \frac{p_{i\,j+1\,k} - p_{i\,j\,k}}{\Delta y}, \qquad \left.\frac{\partial p}{\partial z}\right|_{i\,j\,k+1/2} = \frac{p_{i\,j\,k+1} - p_{i\,j\,k}}{\Delta z_{k+1/2}}, \tag{15}$$



where $\Delta z_{k+1/2} = z_{k+1} - z_k$, and with the obvious modifications to allow for periodic boundary conditions. The divergence of the velocity is needed at $p$-points and is given by

$$\nabla \cdot \boldsymbol{u}|_{ijk} = \frac{u_{i+1/2\,jk} - u_{i-1/2\,jk}}{\Delta x} + \frac{v_{i\,j+1/2\,k} - v_{i\,j-1/2\,k}}{\Delta y} + \frac{w_{ij\,k+1/2} - w_{ij\,k-1/2}}{\Delta z_k}, \tag{16}$$

where $\Delta z_k = z_{k+1/2} - z_{k-1/2}$, with an analogous expression for the divergence of mass flux increments.

Averaging between velocity points and $p$-points is needed at various places in the discretization. Horizontal averaging uses a simple two-point average with weights $1/2$, indicated by $\overline{(.)}^x$ or $\overline{(.)}^y$. For example, the values of $\alpha$ used in the horizontal components of (2), are given by $1/\overline{\rho}^x$ or $1/\overline{\rho}^y$.

Because the vertical grid may be stretched, four different vertical averaging operators are possible:

- $\overline{(.)}^w$: linear interpolation from $p$-levels to $w$-levels;

- $\overline{(.)}^r$: piecewise constant conservative remapping from $p$-levels to $w$-levels;

- $\overline{(.)}^p$: linear interpolation from $w$-levels to $p$-levels;

- $\overline{(.)}^s$: piecewise constant conservative remapping from $w$-levels to $p$-levels.

Complete expressions for these four operators along with some useful conservation and discrete product rule properties are given in Appendix B of Thuburn et al. (2022). For example, the averaging of velocity components to compute the Coriolis terms (equation (2)) and departure points (section 2.6) uses a combination of $\overline{(.)}^x$, $\overline{(.)}^y$, $\overline{(.)}^p$, and $\overline{(.)}^w$. When a value for $p$ is needed at the bottom and top boundaries a variant of the $\overline{(.)}^w$ operator is employed that uses linear extrapolation rather than the default constant extrapolation.

In order to ensure consistent and conservative transport of water and entropy, a key aspect of the model formulation is that a mass budget for a 'dual' $w$-level density $\overline{\rho}^r$ should be satisfied that is consistent with the $p$-level $\rho$ budget (Konor and Arakawa, 2000; Thuburn, 2022; Bendall et al., 2023). Achieving this consistency requires some care in the advection of $w$-level scalars (section 2.6), both for SLICE and for the cheap advection updates, and also in the discrete formulation of surface sources and interior sources (section 2.7).

In the following sections, for clarity, details of the spatial discretization are suppressed except to indicate where different vertical averaging operators are used.

## 2.3 Overview of information flow

Before delving further into details, it is useful to take an overview of the flow of information between the dynamics and thermodynamics parts of the model formulation (Fig. 2), to help clarify the organization of the linearization and the derivation of the Helmholtz problem in subsections 2.8, 2.9 and 2.10.

For the continuous equations, the role of the thermodynamics is to return the pressure $p$, given the density $\rho$, total water $q$, and total entropy $\eta$. To do so, the thermodynamics must determine how $q$ and $\eta$ are partitioned among the different components and phases $q^v$, $q^l$, $q^f$, $\eta^d$, $\eta^v$, $\eta^l$, $\eta^f$.



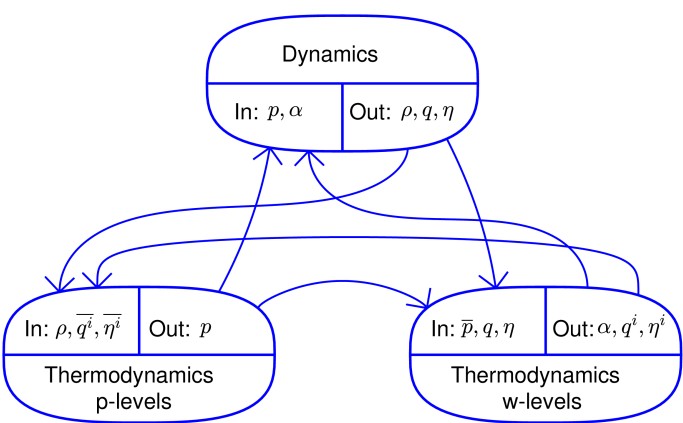

**Figure 2.** Schematic showing the flow of information between the dynamics, thermodynamics at $p$-levels, and thermodynamics at $w$-levels for the case of equilibrium thermodynamics. (For the nonequilibrium case the $w$-level thermodynamics retains some memory of $q^i$ and $\eta^i$.) Superscript $i$ stands for any of d, v, l, or f. An overbar indicates a vertical averaging operation from $p$-levels to $w$-levels or vice-versa.

The situation is more complicated when discretized with a Charney-Phillips vertical grid staggering, since $p$ and $\rho$ are stored at $p$-levels while $q$ and $\eta$ are stored at $w$-levels. Moreover, in order to obtain optimal coupling between vertical velocity and buoyancy, the $w$-level specific volume that appears in the vertical pressure gradient term in (2) must be calculated not simply from a vertically averaged density $1/\overline{\rho}^r$, but from the $w$-level $\eta$ and $q$ and a vertically averaged pressure $\overline{p}^w$ (Thuburn, 2017a). It is convenient to express this requirement through an additional equation

$$\overline{p}^w - p^{(w)} \equiv -R_{\overline{p}} = 0, \tag{17}$$

where $p^{(w)}$ is the pressure appearing in (12) on $w$-levels, and $R_{\overline{p}}$ is a residual that should be driven to zero by the iterative solver. Thus, a full set of thermodynamic calculations (6)-(13) is carried out on $w$-levels, while on $p$-levels the following subset of thermodynamic calculations is carried out:

$$\overline{q^v}^p \alpha^{v\,(p)} + \overline{q^l}^p \alpha^l + \overline{q^f}^p \alpha^f - \frac{1}{\rho} = 0, \tag{18}$$

$$\overline{q^v}^p \alpha^{v\,(p)} - \overline{q^d}^p \alpha^{d\,(p)} = 0, \tag{19}$$

$$p + e_\alpha^d + e_\alpha^v = 0. \tag{20}$$

Equation (18) determines the $p$-level specific volume of water vapour $\alpha^{v\,(p)}$, then (19) determines the $p$-level specific volume of dry air $\alpha^{d\,(p)}$. Finally, the pressure is computed from the dry air and water vapour internal energy potentials in terms of their





$p$-level natural variables $\alpha^{\mathrm{d}\,(p)}$, $\overline{\eta^{\mathrm{d}}}^{p}$, $\alpha^{\mathrm{v}\,(p)}$, and $\overline{\eta^{\mathrm{v}}}^{p}$. Note that the specific volumes $\alpha^{\mathrm{d}\,(p)}$ and $\alpha^{\mathrm{v}\,(p)}$ are calculated directly from $\rho$ on $p$-levels rather than using vertical averages of the $w$-level values $\overline{\alpha^{\mathrm{d}}}^{p}$, $\overline{\alpha^{\mathrm{v}}}^{p}$, ensuring that $p$ responds correctly and locally to changes in $\rho$.

### 2.4 Semi-implicit semi-Lagrangian scheme

Splitting the momentum equation into its horizontal and vertical components with $\boldsymbol{v} = (u, v, 0)$ the horizontal velocity, a semi-implicit semi-Lagrangian discretization of (1)-(5) is

$$[\rho - a\Delta t S_\rho]^{n+1} - [\rho + b\Delta t S_\rho]_{\mathrm{T}}^{n} \equiv -R_\rho = 0, \tag{21}$$

$$[\boldsymbol{v} + a\Delta t\left((2\boldsymbol{\Omega} \times \boldsymbol{u})_{\mathbf{H}} + \alpha\nabla_{\mathbf{H}}p - \boldsymbol{S_v}\right)]^{n+1} - [\boldsymbol{v} - b\Delta t\left((2\boldsymbol{\Omega} \times \boldsymbol{u})_{\mathbf{H}} + \alpha\nabla_{\mathbf{H}}p - \boldsymbol{S_v}\right)]_{\mathrm{D}}^{n} \equiv -\boldsymbol{R_v} = \boldsymbol{0}, \tag{22}$$

$$\left[w + a\Delta t\left((2\boldsymbol{\Omega} \times \boldsymbol{u})_{\mathbf{V}} + \alpha\frac{\partial p}{\partial z} + \frac{\partial \Phi}{\partial z} - S_w\right)\right]^{n+1} - \left[w - b\Delta t\left((2\boldsymbol{\Omega} \times \boldsymbol{u})_{\mathbf{V}} + \alpha\frac{\partial p}{\partial z} + \frac{\partial \Phi}{\partial z} - S_w\right)\right]_{\mathrm{D}}^{n} \equiv -R_w = 0, \tag{23}$$

$$[\overline{\rho}^r\left(\eta - a\Delta t S_\eta\right)]^{n+1} - [\overline{\rho}^r\left(\eta + b\Delta t S_\eta\right)]_{\mathrm{T}}^{n} - \Delta t\overline{\rho}^r \boldsymbol{J}^{\mathrm{T}}\boldsymbol{P}^{n+1} \equiv -R_{\rho\eta} = 0, \tag{24}$$

$$[\overline{\rho}^r\left(q - a\Delta t S_q\right)]^{n+1} - [\overline{\rho}^r\left(q + b\Delta t S_q\right)]_{\mathrm{T}}^{n} \equiv -R_{\rho q} = 0, \tag{25}$$

$$[\boldsymbol{X} - a\Delta t\boldsymbol{S_X}]^{n+1} - [\boldsymbol{X} + b\Delta t\boldsymbol{S_X}]_{\mathrm{D}}^{n} - \Delta t\boldsymbol{J} = \boldsymbol{0}. \tag{26}$$

Here, superscripts $n$ and $n+1$ are time step indices. Subscripts $\mathbf{H}$ and $\mathbf{V}$ indicate horizontal and vertical components, respectively. Subscript D indicates a quantity interpolated or remapped to a semi-Lagrangian trajectory departure point or cell, while subscript T indicates a conservatively transported quantity defined by $\mathcal{V}^{n+1}\psi_{\mathrm{T}} = [\mathcal{V}\psi]_{\mathrm{D}}^{n}$.

$\Delta t$ is the time step, and $a$ and $b = 1 - a$ are off-centring parameters for the dynamics. To damp acoustic waves that might be generated by initial perturbations imposed to trigger turbulence or by the switching on of forcing terms at the initial time, $a$ is smoothly adjusted from 1 to a user specified value over the first 900s of a model run. A value of $a = 0.51$ (after the initial adjustment) is used for the results shown in section 5.

The terms $R_\rho$, $\boldsymbol{R_v}$, $R_w$, $R_{\rho\eta}$, $R_{\rho q}$ represent residuals in their respective equations. In the target discretization these residuals should be zero. The iterative solver described in the following sections attempts to drive those residuals to zero. No residual appears in (26) because that equation is used to diagnose the fluxes $\boldsymbol{J}$ so it is always satisfied exactly.





The subsystem of thermodynamics equations (6)-(13) is solved at step $n+1$, and it is $\boldsymbol{P}^{n+1}$ that appears in (24), effectively giving a backward Euler treatment of the thermodynamics. The thermodynamic processes of interest typically involve relaxation towards equilibrium rather than oscillation about equilibrium. A backward Euler treatment should be sufficiently accurate for nonequilibrium processes whose timescale is much longer than $\Delta t$, such as evaporation of falling rain. For processes whose timescale is shorter than about $\Delta t/2$ a backward Euler step is preferable to a Crank-Nicolson step since Crank-Nicolson can overshoot the equilibrium solution. If the form of the resistivity matrix implies that any part of the system is in equilibrium (e.g. $T^{\mathrm{v}} = T^{\mathrm{d}}$) then a backward Euler step is essential, since the equilibrium must be imposed at step $n+1$, not as a time average.

## 2.5 Time stepping algorithm

The semi-implicit semi-Lagrangian scheme described in section 2.4 is both nonlinear in the unknown step $n+1$ values and nonlocal because unknown values at neighboring gridpoints are coupled. The equations are solved using an iterative quasi-Newton algorithm (algorithm 1). By elimination of unknowns, the linear system for the Newton update is reduced to a more or less standard Helmholtz problem, which is solved using a multigrid method (section 2.11). To avoid the expense of computing conservative semi-Lagrangian transport multiple times per step, a single full advection calculation is made once at the start of the time step, and relatively cheap updates to the transport are made at each solver iteration (Zerroukat and Allen, 2020). These cheap transport updates use simple upwind or centred schemes and are made in such a way that the transport of scalars remains conservative and bounded (assuming conservative and bounded options have been chosen by the user) and consistent with the transport of mass at every solver iteration (Thuburn, 2024).

---

**Algorithm 1** Computations performed to take one model time step

---

Compute time step $n$ terms

Initialize step $n+1$ state variables to step $n$ state

Full advection calculation: compute $[.]_{\mathrm{D}}$ and $[.]_{\mathrm{T}}$ terms

**for** $\ell = 1$ to $N_\ell$ **do**

    Compute step $n+1$ terms based on iteration $\ell-1$ values

    Compute residuals

    Build and solve the Helmholtz problem

    Backsubstitute, updating all transport terms and state variables

**end for**

---

The number of solver iterations takes a default value $N_\ell = 3$. This was found to be sufficient for all test cases simulated except DYCOMS, for which $N_\ell = 4$ was needed. This exceptional case is discussed in sections 4 and 5.

When cheap transport updates are employed, the resulting time integration scheme at solver convergence is not quite as written in section 2.4, with semi-Lagrangian advection by the trajectory-average velocity. Rather, the net transport results from semi-Lagrangian advection using the *first-guess* trajectory-average velocity followed by a sequence of small corrections. Nevertheless, the end result is very close to the target discretization and appears to work well in practice.





An attractive aspect of the use of the consistent and conservative cheap transport updates is that, after the first solver iteration, the residuals in (21), (24) and (25) and the equations for any advected tracers are very small and result only from changes in source terms between one iteration and the next (Thuburn, 2024). It is likely that this helps solver convergence (section 4.4).

## 2.6 Advection

All advection terms are computed using semi-Lagrangian schemes. Different advection options are available depending on the
advected variable and its location on the grid. All momentum equation components are advected with an interpolating (non-conservative) semi-Lagrangian scheme. Terms in the total fluid density equation and the mass fractions of any $p$-level tracers are advected using the mass-conserving SLICE scheme (Zerroukat et al., 2009). Advected terms in the water and entropy equations (24), (25) and (26), and any $w$-level tracers, may be advected with either an interpolating semi-Lagrangian scheme or with SLICE.

Both the interpolating semi-Lagrangian scheme and SLICE use the 'cascade' idea (Purser and Leslie, 1991) to replace a three-dimensional interpolation or remapping by a sequence of one-dimensional interpolations or remappings (Fig. 3).

    The interpolating semi-Lagrangian scheme is based on cubic Lagrange interpolation. For the vertical interpolation, modifications are needed near the upper and lower boundaries. Between the uppermost pair of data points and between the lowermost pair of data points a modified cubic interpolation is used that uses only the two nearest data points and is very close to a
linear interpolation. For terms in the $u$ and $v$ equations, when extrapolation above the uppermost data point is needed constant extrapolation is used. When extrapolation below the lowest data point is needed, the scheme uses either constant extrapolation or linear interpolation between the data value at level 1 and zero at the surface, according to whether the user selects a `'freeslip'` or `'noslip␣␣'` boundary condition for the advection.

    For the SLICE scheme, for each advected field the user may choose between piecewise constant, piecewise parabolic (Colella
and Woodward, 1984) and parabolic spline (Zerroukat et al., 2007) remapping schemes. Experimentation suggests that, for most purposes, piecewise constant remapping is adequate for cell volume, divergence, and density. The results shown in section 5 use this option, with parabolic spline method remapping for entropy and water. For each advected field the user has the option to use a simple limiter ensuring boundedness of the advected field. The limiter is redundant in the case of SLICE advection with piecewise constant remapping. In section 5 the limiter is used for advection of entropy and water but not for advection of
velocity components.

    Some modifications to previous implementations of cascade advection schemes are made to reduce expensive searching and conditional code and to maximise re-use of information. First, in addition to the trajectory departure points, three-dimensional cascade interpolation or remapping requires two sets of 'intermediate departure points'. Rather than construct these intermediate departure points by computing intersections between the arrival coordinate system defined by the model grid and the
departure or Lagrangian coordinate system, as in previous work (e.g Purser and Leslie, 1991; Nair et al., 2002; Zerroukat et al., 2002), here they are computed by separate trajectory calculations. The idea is illustrated in Fig. 3 for the two-dimensional case. A first guess followed by a single fixed point iteration is used for all trajectory calculations.



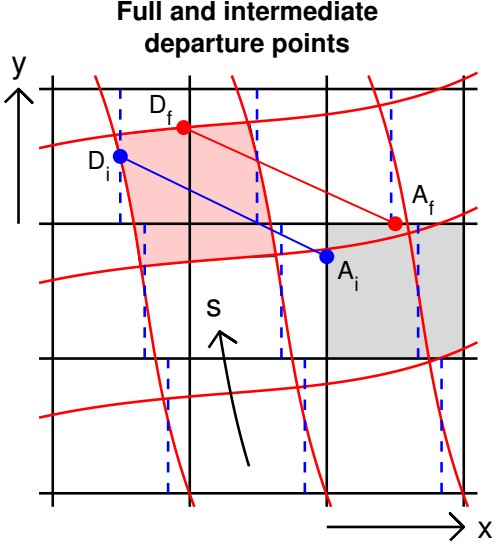

**Figure 3.** Departure points and cascade remapping in two dimensions. The red line $D_f$ to $A_f$ is an example 'full' trajectory to a $v$-point arrival point $A_f$ from the corresponding departure point $D_f$. The $x$- and $y$-coordinates of $A_f$ are known and the $x$ and $y$ coordinates of $D_f$ are to be calculated. The blue line $D_i$ to $A_i$ is an example 'intermediate' trajectory to $A_i$ from $D_i$. The $x$-coordinate of $A_i$ and the $y$-coordinate of $D_i$ are known and the $y$-coordinate of $A_i$ and the $x$-coordinate of $D_i$ are to be calculated. Once the intermediate and full departure points are found, fields are conservatively remapped in the $x$-direction, from the model grid cells (straight black lines) to intermediate cells (blue dashed lines), then in the $s$-direction to the departure cells (red curves). The remapped field is then transported from its departure cells (e.g. pink cell) to the corresponding arrival cells (e.g. grey cell).

Second, we want to remap the density field using a volume coordinate to ensure consistency with the trajectory-average divergence (see below), and we want to remap water, entropy, and tracers in a mass coordinate to ensure that their advection is conservative, bounded if desired, and consistent with the density advection. However, neither the volume coordinate nor the mass coordinate is a simple function of cell index, so it might appear, at first glance, that expensive searching is needed to determine the necessary origin grid indices in these coordinates. This difficulty is circumvented with the aid of the following insight. To compute a one-dimensional remapping of a field $f$ from an origin grid to a destination grid, the information needed comprises the origin grid cell average values of $f$, the origin grid coordinate intervals $\Delta s_k$, the origin grid indices $k_i$ corresponding to destination grid cell edges $i$, and the cell fractions $\xi_i$ in the $s$-coordinate (Fig. 4). Fields are then remapped in the following sequence.

1. Remap cell volume in a geometrical coordinate $x$, $y$ or $z$; the origin grid indices $k_i$ and cell fractions $\xi_i$ are easily determined.





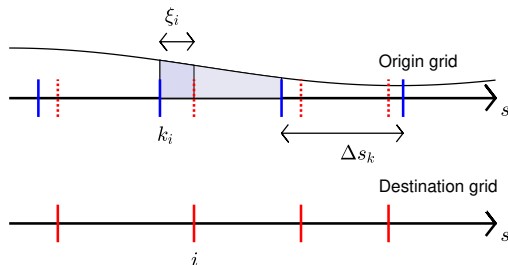

**Figure 4.** Information used for one-dimensional conservative remapping of a field between an origin grid (cell edges indicated in blue) and a destination grid (cell edges indicated in red). The remapping coordinate $s$ may be a geometrical coordinate ($x$, $y$ or $z$), volume, or mass. $\Delta s_k$ is an origin grid cell size in the $s$-coordinate. $k_i$ is the origin grid cell index in which the destination grid edge $i$ sits. $\xi_i$ is the fraction of the origin grid cell $k_i$, measured in the $s$-coordinate, to the left of destination edge $i$.

2. Remap density in a volume coordinate; the origin grid indices $k_i$ are unchanged while the cell fractions $\xi_i$ are obtained as a by-product of the volume remapping. For example, if Fig. 4 represents the remapping of volume in a geometrical coordinate then the volume-coordinate cell fraction needed for the density remapping is given by the ratio of the dark shaded area to the full shaded area.

3. Remap $p$-level tracer mass fractions in a mass coordinate; the origin grid indices $k_i$ are again unchanged while the cell fractions $\xi_i$ are obtained as a by-product of the density remapping.

4. Remap $w$-level scalars in a mass coordinate. The origin grid indices $k_i$ and cell fractions $\xi_i$ must correspond to a mass coordinate based on $\overline{\rho}^r$; they can be computed from the mass coordinate $k_i$ and $\xi_i$ at the $p$-levels immediately below and above the $w$-level in question.

Thuburn et al. (2010) found that the accuracy of a semi-implicit semi-Lagrangian scheme with conservative semi-Lagrangian advection of density relies on the semi-Lagrangian departure volumes (departure areas in the shallow water context) being consistent with the trajectory-average divergence. Subsequent work showed that this divergence-consistency criterion is crucial for stability too, unless a strong off-centring is employed. However, it appears very difficult to enforce this criterion directly within the trajectory calculations. Thuburn et al. (2010) solved this problem by conservatively advecting the divergence field, hence constructing the required departure cell areas, and using an area coordinate for the final SLICE remapping sweep in the advection of mass. Here a slightly different approach is taken, to enable the remapping sequence discussed above. First, the divergence field is conservatively advected, and the information is used to compute the required departure cell volumes. These required departure cell volumes are compared with the actual departure cell volumes returned by SLICE, and the difference is used to compute a (small) divergent velocity increment sufficient to correct the discrepancy. The semi-Lagrangian advection of all other advected fields is then carried out, following which the machinery for making cheap advection updates is used to update the advection of all fields using this velocity increment and so satisfy divergence-consistency.





One of the benefits of the Charney-Phillips vertical grid is that the colocation of $w$ and $\eta$ permits a tight coupling between vertical motion and buoyancy. However, when the entropy transport is computed in flux form that tight coupling is lost unless the horizontal fluxes are corrected to account for the vertical gradients of $\eta$, $F_x = \overline{\rho}^x u$, and $F_y = \overline{\rho}^y v$ (Thuburn, 2022). Thuburn (2022) showed how the horizontal remapping of $w$-level fields in SLICE could be modified to implement the required correction. Here, since the correction is generally small, a much simpler approach is taken. Before carrying out the main SLICE advection of entropy, a small transport correction is made to the entropy using the horizontal fluxes

$$
F_x^\eta = a^w b^w \Delta z^2 \overline{\frac{\partial F_x}{\partial z} \overline{\frac{\partial \eta}{\partial x}}^x}^w , \qquad F_y^\eta = a^w b^w \Delta z^2 \overline{\frac{\partial F_y}{\partial z} \overline{\frac{\partial \eta}{\partial y}}^y}^w , \tag{27}
$$

where $a^w$ and $b^w$ are the coefficients associated with the $\overline{(.)}^w$ operator. All quantities are evaluated at step $n$, and the fluxes are applied over the time step $\Delta t$. Subscripts $k+1/2$ indicating the vertical level have been omitted for clarity. Equation (27) is a slight generalization of equation (7) of Thuburn (2022) to allow for a vertically stretched grid. An analogous correction is applied to the transport of all conservatively transported $w$-level scalars. However, because this simple formulation does not guarantee boundedness of advected scalars, the user may choose whether or not to include the correction. The correction is included for the results shown in section 5.

In the backsubstitution stage of the time step cheap updates are made to all transported terms $[.]_\mathrm{D}$ and $[.]_\mathrm{T}$ to account for the velocity increments $\boldsymbol{u}'$, applied over the backward part of the time step $a\Delta t$. The transported terms in the momentum equation are updated using an advective-form first-order upwind scheme. The transported term in the density equation is updated using a flux-form scheme with mass flux increments $\boldsymbol{F}' = \hat{\rho}\boldsymbol{u}'$ where the cell edge values $\hat{\rho}$ are given by a second-order centred scheme. The transported terms in any $p$-level tracer equations are updated using a flux-form scheme with tracer flux increments $\boldsymbol{F}^{\chi\prime} = \boldsymbol{F}'\hat{\chi}$ where the cell edge values $\hat{\chi}$ are given by a first-order upwind scheme. Finally, the transport terms for any $w$-level scalars are updated using a flux-form scheme with scalar flux increments $\boldsymbol{F}^{\chi\prime} = \overline{\boldsymbol{F}'}^d \hat{\chi}$ with $\hat{\chi}$ again given by a first-order upwind scheme. The notation $\overline{(.)}^d$ here indicates that horizontal mass flux increments have been mapped to $w$-levels using the $\overline{(.)}^r$ operator and vertical mass flux increments have been mapped to $p$-levels using the $\overline{(.)}^p$ operator for consistency with the dual mass budget (Thuburn et al., 2022). This way of constructing the scalar flux increments ensures that the scalar transport remains consistent with density transport and conservative and bounded if the full semi-Lagrangian transport was conservative and bounded.

## 2.7 Boundary and interior forcing terms

Each simulated case requires the implementation of appropriate boundary and interior forcing terms. This section highlights some aspects of how those terms are implemented for consistency with the rest of the model formulation.

### 2.7.1 Consistent surface fluxes of mass and water

Under a careful and consistent treatment of the moist thermodynamics, a surface flux of water implies a surface flux of mass. Most atmospheric models neglect that flux of mass. Because of the Charney-Phillips vertical staggering, and the fact that the





predicted density is the total density rather than the dry density, care is needed in computing the density and water increments to ensure that the changes in both total mass and total water within the model agree with the time-integrated surface fluxes.

First consider the forward part of the time step, i.e., the $[\ldots]^n$ terms in (21) and (25). Let $\mathcal{F}^\rho$ be the surface flux of mass and $\overline{\rho^n}^r_{1/2}\mathcal{F}^q$ be the surface flux of water (both in $\mathrm{kg\,m^{-2}s^{-1}}$). Recall that the $w$-level density at level $k+1/2$ must equal the

conservatively remapped $p$-level density $\overline{\rho}^r_{k+1/2}$. Thus, any change in total density at level 1 affects the density of water at level $3/2$, $\overline{\rho^n}^r_{3/2}q_{3/2}$, as well as the density of water at level $1/2$, $\overline{\rho^n}^r_{1/2}q_{1/2}$. Nevertheless, imposing the constraints

– only the density at level 1 may change, and the total change in column mass per unit area must equal $\mathcal{F}^\rho b\Delta t$,

– only the specific humidity at level $1/2$ may change, and the total change in column water mass per unit area must equal $\overline{\rho^n}^r_{1/2}\mathcal{F}^q b\Delta t$,

leads to unique solutions for the increment in density at level 1

$$\delta\rho_1 = \frac{\mathcal{F}^\rho b\Delta t}{\Delta z_1} \tag{28}$$

and the increment in specific humidity at level $1/2$

$$\delta q_{1/2} = \frac{\left(\rho_1^n\mathcal{F}^q - \overline{q^n}^s_1\mathcal{F}^\rho\right)b\Delta t}{\rho_1^+\Delta z_{1/2}}, \tag{29}$$

where $\rho_1^+ = \rho_1^n + \delta\rho_1$, and we have used the fact that $\overline{\rho}^r_{1/2} \equiv \rho_1$.

For the backward part of the timestep, i.e., the $[\ldots]^{n+1}$ terms in (21) and (25), an analogous argument leads to

$$\delta\rho_1 = \frac{\mathcal{F}^\rho a\Delta t}{\Delta z_1} \tag{30}$$

and

$$\delta q_{1/2} = \frac{\left(\rho_1^{n+1}\mathcal{F}^q - \overline{q^{n+1}}^s_1\mathcal{F}^\rho\right)a\Delta t}{\rho_1^-\Delta z_{1/2}}, \tag{31}$$

where $\rho_1^- = \rho_1^{n+1} - \delta\rho_1$. Terms at time step $n+1$ are approximated by the latest available estimate.

The treatment of surface entropy fluxes is analogous, with $q$ replaced by $\eta$ and $\mathcal{F}^q$ replaced by $\mathcal{F}^\eta$.

### 2.7.2 Surface drag

The various test cases implemented specify how the vertical flux of horizontal momentum is to be parameterized at the surface. Since there are no parameterized subgrid fluxes in the interior of the domain, the convergence of the parameterized momentum flux occurs entirely in the lowest model layer, i.e., it is applied to the $u$ and $v$ components at model level 1.

Some of the test cases implemented specify that the surface momentum flux is to be computed using Monin-Obukhov similarity theory. However, there are two difficulties. First, Monin-Obukhov similarity theory expresses the mean flow speed $U(z)$ at some height $z$ as a nonlinear function of the friction velocity $U_*$. What we require in a numerical model is to express





the friction velocity, hence the surface momentum flux, in terms of the known flow speed at the lowest model level; however, in the usual approach, this requires the iterative solution of a nonlinear problem in each grid column. Second, and more
seriously, in the stable case (negative surface buoyancy flux) in light winds Monin-Obukhov similarity theory is valid only over a limited height range that might be less than the height of the lowest model level $z_1$. In this case, with commonly used stability functions, the theory can produce either no solution or multiple solutions for $U_*$, given $U(z_1)$. To avoid these difficulties, a slight reformulation of Monin-Obukhov theory is used to compute surface momentum fluxes (Appendix A4).

### 2.7.3 Consistent interior forcing terms

For test cases that require a source of moisture in the interior of the model domain, consistency between the $p$-level mass budget and the $w$-level dual mass budget requires that

$$(\rho S_q)_{k+1/2} = \left(\overline{S_\rho}^r\right)_{k+1/2}. \tag{32}$$

The simplest way to satisfy this condition is to specify the required moisture/mass source first on $p$-levels, and then remap conservatively to $w$-levels. For simplicity, interior entropy sources are specified in the same way.

## 2.8 Linearized dynamics

To take a model time step, a quasi-Newton method is used to solve equations (21)-(26) along with the $p$-level and $w$-level thermodynamic equations. After some number of quasi-Newton iterations, using the latest available estimates to evaluate step $n+1$ terms, the residuals $R_\rho$, $\boldsymbol{R_v}$, $R_w$, $R_{\rho\eta}$, $R_{\rho q}$, $R_{\rm PE}$, $R_{\overline{p}}$ will generally be non-zero. We seek increments or updates to the step $n+1$ model variables that will reduce those residuals. This is done through an approximate linearization of the governing
equations (this section and section 2.9), leading to a large linear system for the increments. Despite the apparent complexity of this system, systematic elimination of unknowns, partly manually and partly numerically, leads to a nearly standard Helmholtz problem for a single unknown per model gridpoint: the pressure increment $p'$ (section 2.10). The Helmholtz problem can be solved efficiently using well-established methods; here a multigrid method is used. Once $p'$ is found, the other increments can be found through backsubstitution (section 2.11).
The approximate linearization of (21)-(26) is

$$\rho' + a\Delta t \nabla \cdot \boldsymbol{F'} = R_\rho, \tag{33}$$

$$\boldsymbol{v'} + a\Delta t \left(\alpha_* \nabla_{\mathbf{H}} p' - \boldsymbol{S'_v}\right) = \boldsymbol{R_v}, \tag{34}$$

$$w' + a\Delta t \left(\alpha_* \frac{\partial p'}{\partial z} + \alpha' \frac{\partial p_*}{\partial z}\right) = R_w, \tag{35}$$





$$\overline{\rho^{(\ell+1)}}^{r}\eta' + \overline{\rho'}^{r}\left(\eta - a\Delta t S_\eta\right)^{(\ell)} - \left[\overline{\rho}^{r}\left(\eta + b\Delta t S_\eta\right)\right]'_{\mathrm{T}} = R_{\rho\eta}, \tag{36}$$

$$\overline{\rho^{(\ell+1)}}^{r}q' + \overline{\rho'}^{r}\left(q - a\Delta t S_q\right)^{(\ell)} - \left[\overline{\rho}^{r}\left(q + b\Delta t S_q\right)\right]'_{\mathrm{T}} = R_{\rho q}, \tag{37}$$

$$\boldsymbol{X}' - \Delta t \boldsymbol{J}' = 0. \tag{38}$$

An asterisk on any variable indicates a reference value for the linearization. Here the most recent estimate for the step $n+1$ value is used as the reference value, avoiding the need to store additional three-dimensional fields. Certain terms in (36) and (37) are explicitly evaluated at iteration number $\ell$ or $\ell+1$. This specific way of writing the linearization ensures that, when

(36) and (37) are used in the backsubstitution, $\eta$ and $q$ are incremented exactly according to the consistent and conservative transport updates. Since the source terms $S_\rho$ etc. are generally non-stiff, their linearizations are mostly omitted. The exception is a linearization of the surface drag at model level 1 $\boldsymbol{S}'_{\boldsymbol{v}} = -\boldsymbol{v}'/\tau_{\mathrm{drag}}$, where $\tau_{\mathrm{drag}}$ is a surface drag timescale computed alongside the surface momentum flux. This term is omitted from the Helmholtz problem (in principle it could be included) but is included in the backsubstitution.

In order to arrive at a standard Helmholtz problem for the pressure increments we must make a $w'N^2$ term appear in the linearization, where $N^2$ is an appropriately defined buoyancy frequency squared. This requires us to work with a linearization of advective form transport equations for $\eta$ and $q$. Consider the $\eta$ equation (36). In the backsubstitution the transport increment is computed as

$$\left[\overline{\rho}^{r}\left(\eta + b\Delta t S_\eta\right)\right]'_{\mathrm{T}} = -a\Delta t \nabla \cdot \left(\overline{\boldsymbol{F}'}^{d}\hat{\eta}\right) \tag{39}$$

for some (upwind) cell edge values $\hat{\eta}$. Thus, (36) becomes

$$\overline{\rho^{(\ell+1)}}^{r}\eta' + \overline{\rho'}^{r}\left(\eta - a\Delta t S_\eta\right)^{(\ell)} + a\Delta t \nabla \cdot \left(\overline{\boldsymbol{F}'}^{d}\hat{\eta}\right) = R_{\rho\eta}. \tag{40}$$

Subtracting $\hat{\eta}$ times a vertical average of (33) and approximating $\overline{\rho^{(\ell+1)}}^{r}$ by a generic reference density $\rho_*$ gives

$$\eta' + a\Delta t w' \frac{\partial \eta}{\partial z} = \frac{R_{\rho\eta} - \hat{\eta}\overline{R_\rho}}{\rho_*} \equiv R_\eta. \tag{41}$$

Proceeding in a similar way for the $q$ equation (37) gives

$$q' + a\Delta t w' \frac{\partial q}{\partial z} = \frac{R_{\rho q} - \hat{q}\overline{R_\rho}}{\rho_*} \equiv R_q. \tag{42}$$





## 2.9 Linearized thermodynamics

When the $w$-level thermodynamic state is updated, certain fields are diagnosed, ensuring that (6)-(11) are satisfied exactly: $q^{\mathrm{d}}$ is set equal to $1-q$; (9) gives $q^{\mathrm{v}}$; (10) gives $\alpha^{\mathrm{v}}$; and (6) gives $\alpha^{\mathrm{d}}$. Also, the updating of terms in (7) and (8) during backsubstitution is done in such a way that those equations remain exactly satisfied. Thus, no residual term appears in the linearized versions of these equations.

The linearized forms of (6)-(11) are then as follows:

$$q^{\mathrm{v}}\alpha^{\mathrm{v}\prime} + \alpha^{\mathrm{v}}q^{\mathrm{v}\prime} - q^{\mathrm{d}}\alpha^{\mathrm{d}\prime} + \alpha^{\mathrm{d}}q' = 0, \tag{43}$$

$$(1-\delta^{\mathrm{l}})\lambda^{\mathrm{l}\prime} - \delta^{\mathrm{l}}q^{\mathrm{l}\prime} = 0, \tag{44}$$

$$(1-\delta^{\mathrm{f}})\lambda^{\mathrm{f}\prime} - \delta^{\mathrm{f}}q^{\mathrm{f}\prime} = 0, \tag{45}$$

$$q^{\mathrm{v}\prime} + q^{\mathrm{l}\prime} + q^{\mathrm{f}\prime} - q' = 0, \tag{46}$$

$$q^{\mathrm{v}}\alpha^{\mathrm{v}\prime} + \alpha^{\mathrm{v}}q^{\mathrm{v}\prime} + \alpha^{\mathrm{l}}q^{\mathrm{l}\prime} + \alpha^{\mathrm{f}}q^{\mathrm{f}\prime} - \alpha' = 0, \tag{47}$$

$$q^{\mathrm{d}}\eta^{\mathrm{d}\prime} - \eta^{\mathrm{d}}q' + q^{\mathrm{v}}\eta^{\mathrm{v}\prime} + \eta^{\mathrm{v}}q^{\mathrm{v}\prime} + q^{\mathrm{l}}\eta^{\mathrm{l}\prime} + \eta^{\mathrm{l}}q^{\mathrm{l}\prime} + q^{\mathrm{f}}\eta^{\mathrm{f}\prime} + \eta^{\mathrm{f}}q^{\mathrm{f}\prime} - \eta' = 0. \tag{48}$$

Note that $q^{\mathrm{d}\prime} = -q'$ has been used to eliminate increments of the dry mass fraction. The linearization of (17) is

$$\overline{p'}^{w} - p^{(w)\prime} = R_{\overline{p}}. \tag{49}$$

With the aid of (38), the linearized phenomenological equations become

$$\frac{1}{\Delta t}\mathbf{R}\boldsymbol{X}' - \boldsymbol{P}' + \boldsymbol{C}' = \boldsymbol{R}_{\mathrm{PE}}; \tag{50}$$

for brevity the details of $\boldsymbol{P}'$ and $\boldsymbol{C}'$ are suppressed.

Two modifications are then made to the linear system. First, a change of variable is made, introducing $\tilde{\lambda}^{\mathrm{l}\prime} = \lambda^{\mathrm{l}\prime} + q^{\mathrm{l}\prime}$ and $\tilde{\lambda}^{\mathrm{f}\prime} = \lambda^{\mathrm{f}\prime} + q^{\mathrm{f}\prime}$. Second, $\eta_{\mathrm{ref}}$ times (44) is added to the $q^{\mathrm{l}\prime}$ phenomenological equation and $\eta_{\mathrm{ref}}$ times (45) is added to the $q^{\mathrm{f}\prime}$ phenomenological equation. These two modifications guarantee that the coefficient of $q^{\mathrm{l}\prime}$ in (44) and the coefficient of $q^{\mathrm{f}\prime}$ in (45) are nonzero, and that the coefficients of $\tilde{\lambda}^{\mathrm{l}\prime}$ and $\tilde{\lambda}^{\mathrm{f}\prime}$ in the relevant phenomenological equations are nonzero. In this way,





the sparsity pattern of the system matrix is known irrespective of the state of the switches $\delta^{\mathrm{l}}$, $\delta^{\mathrm{f}}$, and is then effectively the same at all grid points.

The resulting system of linear equations may be compactly written

$$\widetilde{\mathbf{M}}Z' = \widetilde{\boldsymbol{R}}_{\mathbf{M}}, \tag{51}$$

where $\widetilde{\mathbf{M}}$ is an $11 \times 14$ matrix, and $(Z')^{\mathrm{T}} = ((\boldsymbol{Y}')^{\mathrm{T}}, q', \alpha', \eta')$ where $(\boldsymbol{Y}')^{\mathrm{T}} = \left(\alpha^{\mathrm{v}'}, q^{\mathrm{l}'}, q^{\mathrm{f}'}, q^{\mathrm{v}'}, \alpha^{\mathrm{d}'}, \eta^{\mathrm{d}'}, \eta^{\mathrm{v}'}, \eta^{\mathrm{l}'}, \eta^{\mathrm{f}'}, \tilde{\lambda}^{\mathrm{l}'}, \tilde{\lambda}^{\mathrm{f}'}\right)$. The different ordering of the rows and columns of $\widetilde{\mathbf{M}}$ compared to Bowen and Thuburn (2022a) allows a better exploitation of the sparsity in the Gaussian elimination step discussed in section 2.10.

The linearized versions of the $p$-level thermodynamic equations (18)-(20) are

$$\overline{q^{\mathrm{v}\prime}}^p \alpha^{\mathrm{v}\,(p)} + \overline{q^{\mathrm{v}}}^p \alpha^{\mathrm{v}\,(p)\prime} + \overline{q^{\mathrm{l}\prime}}^p \alpha^{\mathrm{l}} + \overline{q^{\mathrm{f}\prime}}^p \alpha^{\mathrm{f}} + \frac{\rho'}{\rho_*^2} = 0, \tag{52}$$

$$\overline{q^{\mathrm{v}\prime}}^p \alpha^{\mathrm{v}\,(p)} + \overline{q^{\mathrm{v}}}^p \alpha^{\mathrm{v}\,(p)\prime} + \overline{q^{\prime}}^p \alpha^{\mathrm{d}\,(p)} - \overline{q^{\mathrm{d}}}^p \alpha^{\mathrm{d}\,(p)\prime} = 0, \tag{53}$$

$$p' + e_{\alpha\alpha}^{\mathrm{d}} \alpha^{\mathrm{d}\,(p)\prime} + e_{\alpha\eta}^{\mathrm{d}} \overline{\eta^{\mathrm{d}\prime}}^p + e_{\alpha\alpha}^{\mathrm{v}} \alpha^{\mathrm{v}\,(p)\prime} + e_{\alpha\eta}^{\mathrm{v}} \overline{\eta^{\mathrm{v}\prime}}^p = 0. \tag{54}$$

## 2.10 Derivation of the Helmholtz problem

In order to derive a Helmholtz problem, it will be useful to express all other thermodynamic increments at $w$-levels (the components of $\boldsymbol{Y}'$) in terms of $q'$, $\alpha'$, and $\eta'$. This is done by carrying out a numerical Gaussian elimination on $\widetilde{\mathbf{M}}$ to leave

$$\mathbf{M}Z' = \boldsymbol{R}_{\mathbf{M}}, \tag{55}$$

where $\mathbf{M}$ is of the form

$$\mathbf{M} = (\mathbf{I} \; \boldsymbol{C}_1 \; \boldsymbol{C}_2 \; \boldsymbol{C}_3), \tag{56}$$

$\mathbf{I}$ is the $11 \times 11$ identity matrix, and $\boldsymbol{C}_1$, $\boldsymbol{C}_2$, and $\boldsymbol{C}_3$ are columns of (generally) nonzero entries. For efficiency, the Gaussian elimination exploits the known sparsity pattern of the matrix $\widetilde{\mathbf{M}}$ and, since the sparsity pattern is the same for all grid points, the elimination can be done without conditional code, and can be implemented with the innermost loop over model levels.

The entries $M_{ij}$ of the eliminated matrix $\mathbf{M}$ and the entries $R_{\mathbf{M}i}$ of the eliminated right hand side $\boldsymbol{R}_{\mathbf{M}}$ are used in building the coefficients and right hand side of the Helmholtz problem. A subset of them (rows 2, 3, and 7–11 of $\boldsymbol{C}_1$, $\boldsymbol{C}_2$, $\boldsymbol{C}_3$, and $\boldsymbol{R}_{\mathbf{M}}$, corresponding to the equations for $(\lambda^{\mathrm{l}'}, \lambda^{\mathrm{f}'}, q^{\mathrm{l}'}, q^{\mathrm{f}'}, \eta^{\mathrm{v}'}, \eta^{\mathrm{l}'}, \eta^{\mathrm{f}'})$) are saved for use in backsubstitution.

Next, we need an equation relating $p'$, $\alpha'$ and $w'$ at $w$-levels and an analogous equation relating $p'$, $\rho'$ and $w'$ at $p$-levels. With the aid of (55), the pressure perturbation at $w$-levels is given by

$$
\begin{aligned}
p^{(w)\prime} &= -e_{\alpha\alpha}^{\mathrm{d}} \alpha^{\mathrm{d}\prime} - e_{\alpha\eta}^{\mathrm{d}} \eta^{\mathrm{d}\prime} - e_{\alpha\alpha}^{\mathrm{v}} \alpha^{\mathrm{v}\prime} - e_{\alpha\eta}^{\mathrm{v}} \eta^{\mathrm{v}\prime} \\
&= \frac{\partial p}{\partial q} q' + \frac{\partial p}{\partial \alpha} \alpha' + \frac{\partial p}{\partial \eta} \eta' + R_{\mathrm{TDP}},
\end{aligned} \tag{57}
$$





where

$$\frac{\partial p}{\partial q} = e^{\mathrm{d}}_{\alpha\,\alpha} M_{5\,12} + e^{\mathrm{d}}_{\alpha\,\eta} M_{6\,12} + e^{\mathrm{v}}_{\alpha\,\alpha} M_{1\,12} + e^{\mathrm{v}}_{\alpha\,\eta} M_{7\,12}, \tag{58}$$

$$\frac{\partial p}{\partial \alpha} = e^{\mathrm{d}}_{\alpha\,\alpha} M_{5\,13} + e^{\mathrm{d}}_{\alpha\,\eta} M_{6\,13} + e^{\mathrm{v}}_{\alpha\,\alpha} M_{1\,13} + e^{\mathrm{v}}_{\alpha\,\eta} M_{7\,13}, \tag{59}$$

$$\frac{\partial p}{\partial \eta} = e^{\mathrm{d}}_{\alpha\,\alpha} M_{5\,14} + e^{\mathrm{d}}_{\alpha\,\eta} M_{6\,14} + e^{\mathrm{v}}_{\alpha\,\alpha} M_{1\,14} + e^{\mathrm{v}}_{\alpha\,\eta} M_{7\,14}, \tag{60}$$

$$R_{\mathrm{TDP}} = e^{\mathrm{d}}_{\alpha\,\alpha} R_{\mathbf{M}5} + e^{\mathrm{d}}_{\alpha\,\eta} R_{\mathbf{M}6} + e^{\mathrm{v}}_{\alpha\,\alpha} R_{\mathbf{M}1} + e^{\mathrm{v}}_{\alpha\,\eta} R_{\mathbf{M}7}. \tag{61}$$

Defining the sound speed $c$ by

$$\frac{\partial p}{\partial \alpha} = -\rho_*^2 c^2 \tag{62}$$

and using (36) and (37) to eliminate $\eta'$ and $q'$, (49) becomes

$$\frac{\overline{p'}^{\,w}}{c^2} + \rho_*^2 \alpha' + a\Delta t \rho_* w' \frac{N^2}{\mathrm{g}} = R_{\mathrm{buoy}}, \tag{63}$$

where

$$\rho_* \frac{N^2}{\mathrm{g}} = \frac{1}{c^2}\left\{ \frac{\partial p}{\partial q}\frac{\partial q}{\partial z} + \frac{\partial p}{\partial \eta}\frac{\partial \eta}{\partial z} \right\} \tag{64}$$

and

$$R_{\mathrm{buoy}} = \frac{1}{c^2}\left\{ R_{\overline{p}} + R_{\mathrm{TDP}} + \frac{\partial p}{\partial q} R_q + \frac{\partial p}{\partial \eta} R_\eta \right\}. \tag{65}$$

Deriving the analogous equation on $p$-levels is a little more subtle because $p'$ depends both on $\rho'$ and, via $q^{i\,\prime}$ and $\eta^{i\,\prime}$, on the $w$-level $\alpha'$ averaged to $p$-levels. Setting $q^{i\,\prime}$ and $\eta^{i\,\prime}$ to zero in (52)-(54) shows that

$$\left.\frac{\partial p}{\partial \rho}\right|_{\overline{\alpha}^p} = \frac{1}{\rho_*^2}\left( \frac{e^{\mathrm{d}}_{\alpha\,\alpha}}{\overline{q^{\mathrm{d}}}^p} + \frac{e^{\mathrm{v}}_{\alpha\,\alpha}}{\overline{q^{\mathrm{v}}}^p} \right) \equiv \hat{c}^2, \tag{66}$$

where $\hat{c}$ is a 'reduced sound speed' (typically very close to and slightly larger than $c$). Hence

$$\left.\frac{\partial p}{\partial \overline{\alpha}^p}\right|_\rho = -\rho_*^2(c^2 - \hat{c}^2), \tag{67}$$

while $\partial p/\partial q$ and $\partial p/\partial \eta$ are given sufficiently accurately by their $w$-level values averaged to $p$-levels. Proceeding in this way, (54) becomes

$$p' - \hat{c}^2\rho' + \rho_*^2(c^2 - \hat{c}^2)\overline{\alpha'}^p - \frac{\partial p}{\partial q}\overline{q'}^p - \frac{\partial p}{\partial \eta}\overline{\eta'}^p = \overline{R_{\mathrm{TDP}}}^p. \tag{68}$$

Using (63) to eliminate $\alpha'$ and (36) and (37) to eliminate $\eta'$ and $q'$ leaves

$$\frac{p'}{\hat{c}^2} - \rho' + \left( \frac{1}{c^2} - \frac{1}{\hat{c}^2} \right)\overline{\overline{p'}^{\,w}}^{\,p} + a\Delta t \overline{\rho_* w' \frac{N^2}{\mathrm{g}}}^{\,p} = \overline{R_{\mathrm{buoy}}}^{\,p} - \frac{\overline{R_{\overline{p}}}^{\,p}}{\hat{c}^2}. \tag{69}$$





Next, use (63) to eliminate $\alpha'$ from (35) and combine the terms involving $p'$ into a single operator:

$$585 \quad \rho_* w' + \mathcal{D}_1(p') = \frac{\rho_* R_w + a\Delta t \mathrm{g} R_{\mathrm{buoy}}}{1 + a^2\Delta t^2 N^2} \equiv \rho_* R_{wDp}, \tag{70}$$

where

$$\mathcal{D}_1(p') \equiv \frac{a\Delta t}{1 + a^2\Delta t^2 N^2}\left(\frac{\partial p'}{\partial z} + \frac{\mathrm{g}}{c^2}\overline{p'}^{\,w}\right). \tag{71}$$

Also, use (69) to eliminate $\rho'$ from (33) and combine the terms involving $w'$ into a single operator:

$$\frac{p'}{\hat{c}^2} + \left(\frac{1}{c^2} - \frac{1}{\hat{c}^2}\right)\overline{\overline{p'}^{\,w}}^{\,p} + a\Delta t\nabla_{\mathbf{H}}\cdot(\rho_*\boldsymbol{v}') + \mathcal{D}_2(\rho_*w') = \overline{R_{\mathrm{buoy}}}^{\,p} - \frac{\overline{R_{\overline{p}}}^{\,p}}{\hat{c}^2} + R_\rho, \tag{72}$$

where

$$\mathcal{D}_2(\rho_*w') \equiv a\Delta t\left(\frac{\partial}{\partial z}\rho_*w' + \frac{N^2}{\mathrm{g}}\overline{\rho_*w'}^{\,p}\right). \tag{73}$$

Finally, eliminate $u'$, $v'$, and $w'$ using (34) and (70) to obtain the Helmholtz problem

$$\frac{p'}{\hat{c}^2} + \left(\frac{1}{c^2} - \frac{1}{\hat{c}^2}\right)\overline{\overline{p'}^{\,w}}^{\,p} - a^2\Delta t^2\nabla_{\mathbf{H}}\cdot\nabla_{\mathbf{H}}p' - \mathcal{D}_2\mathcal{D}_1(p') = \overline{R_{\mathrm{buoy}}}^{\,p} - \frac{\overline{R_{\overline{p}}}^{\,p}}{\hat{c}^2} + R_\rho - a\Delta t\nabla_{\mathbf{H}}\cdot\boldsymbol{R_v} - \mathcal{D}_2(\rho_*R_{wDp}). \tag{74}$$

The form of the Helmholtz problem is slightly unusual in the appearance of the first two terms rather than a single term
$p'/c^2$. As noted above, the $\overline{\overline{p'}^{\,w}}^{\,p}$ term appears because of the dependence of $p'$ on $\overline{\alpha'}^{\,p}$ and of $\alpha'$ on $\overline{p'}^{\,w}$. Nevertheless, this slightly different form does not affect the stencil of the discrete Helmholtz operator or the difficulty of solving the Helmholtz equation numerically.

The vertical part of the Helmholtz operator must be modified near the top and bottom boundaries to impose $w' = 0$ there. This modification amounts to

– omitting the contributions from levels $1/2$ and $N_z + 1/2$ in $\mathcal{D}_2(\rho_*R_{wDp})$ on the right hand side of (74);

    – omitting the contributions $\mathcal{D}_1(p')$ from levels $1/2$ and $N_z + 1/2$ in $\mathcal{D}_2\mathcal{D}_1(p')$.

The switching of the Lagrange multipliers used to enforce non-negativity of $q^{\mathrm{l}}$ and $q^{\mathrm{f}}$ can significantly change the linearization of the thermodynamics, particularly the value of $N^2$. Therefore, the matrix $\mathbf{M}$ and the coefficients of the Helmholtz problem are rebuilt at every solver iteration.

## 2.11 Helmholtz solver and backsubstitution

The Helmholtz problem is solved using a horizontal multigrid method. Each smoother iteration uses a Jacobi method in the horizontal with a tridiagonal direct vertical solve. Key parameters of the multigrid solver—the depth of the V-cycles, the number of smoother iterations on the coarsest grid, and the number of V-cycles—are automatically chosen to ensure that the pressure increments are sufficiently accurate (Appendix A3).

Having found the solution for $p'$, increments to other variables are found through backsubstitution:





– $u'$, $v'$ and $w'$ are found using (34) and (70);

– Velocity increments are used to compute mass flux increments $\boldsymbol{F}'$ and hence $\rho'$ using (33);

– Velocity and mass flux increments are used to update all transported terms and hence increment $q$ and $\eta$;

– $\alpha'$ is computed using (63), then $(\lambda^{l'}, \lambda^{f'}, q^{l'}, q^{f'}, \eta^{v'}, \eta^{l'}, \eta^{f'})$, are computed from $q'$, $\eta'$ and $\alpha'$ using the coefficients saved after the Gaussian elimination (55).

Before applying the increments $(\lambda^{l'}, \lambda^{f'}, q^{l'}, q^{f'}, \eta^{v'}, \eta^{l'}, \eta^{f'})$, a check is made to see if they would imply breaking any of the constraints $q^l \geq 0$, $q^f \geq 0$, $\lambda^l \leq 0$, $\lambda^f \leq 0$. If they would, then, *at that grid point*, a partial increment is made to these seven variables such that the updated variables sit at the constraint boundary, and the corresponding $\delta^i$ is switched from 0 to 1 or vice versa. All other variables receive their full increments.

After the completion of the solver iterations the model's pressure field is diagnosed using (18)-(20). This ensures that the pressure is consistent with the other model thermodynamic fields and enables bit-reproducible restarts without the need to save the pressure field. During early stages of model development (18)-(20) were also used to recompute $p$ at the end of each solver iteration. However, this formulation is adversely impacted by the switching of constraints, as follows. The total entropy always receives its full increment $\eta'$. When constraint switching leads to partial increments $\eta^{v'}$, $\eta^{l'}$ and $\eta^{f'}$, the dry entropy must take up the slack, potentially resulting in pressure changes much larger than the $p'$ originally returned by the Helmholtz problem, and hence to large residuals locally in the momentum equation. Therefore, in the present formulation, $p$ is updated by adding the increment $p'$ returned by the Helmholtz problem at each solver iteration. In the presence of constraint switching, this alternative way of updating the pressure greatly reduces the residuals in the momentum equation but leads to a larger residual $R_{\overline{p}}$ in (17). Nevertheless, the overall effect is beneficial, accelerating solver convergence.

## 3 Test cases

### 3.1 Generating initial data;

The model includes general purpose routines for constructing a horizontally uniform initial state in discrete hydrostatic balance given specified vertical profiles of a temperature variable, a humidity variable, and the horizontal wind components. The temperature variable may be potential temperature, liquid water potential temperature, virtual potential temperature, or temperature itself. The humidity variable may be total specific humidity or relative humidity.

PTerodaC³TILES version 1.0 has routines in place, which can be selected via namelist options, to generate initial data and forcing terms for the following test cases: ATEX (Stevens et al., 2001); ARM (Brown et al., 2002); BOMEX (Siebesma et al., 2003); BUBBLE (a 3D version of Bryan and Fritsch, 2002); CBL (a dry convective boundary layer, Sullivan and Patton, 2011); DYCOMS (Stevens et al., 2005); a NEUTRAL boundary layer with shear. The user can easily implement new test cases by using exising routines as templates.





PTerodaC$^3$TILES version 1.0 can also generate initial data for the LBA case (Grabowski et al., 2006). However, the current version cannot represent the microphysics and precipitation necessary to simulate this case successfully, and the forcing terms are not yet implemented.

### 3.2 Output and diagnostics

A variety of diagnostics are computed online and may be selected by the user for output. These include

- time series of global diagnostics such as total mass and total water, along with accumulated source terms;

- column diagnostics of horizontal means of model fields and derived quantities such as turbulent kinetic energy, vertical fluxes, and cloud fraction;

- two dimensional slices in the $x$-$y$, $x$-$z$, and $y$-$z$ planes of the main model fields plus some derived quantities like cloud 650 top height;

- diagnostics of quasi-Newton solver convergence;

- diagnostics of quantities potentially related to model stability.

Full details are given in the User Manual, available from Zenodo; see the *Code availability* section.

## 4 Verification

All components of the model have been thoroughly tested during development. This section highlights some aspects that require particularly careful checking or that can appear to work despite errors in formulation or coding.

### 4.1 Advection

The advection routines have been tested with specified velocity fields, independently of the rest of the model, to verify their overall accuracy, including correct behaviour at vertical and lateral boundaries and, where relevant, to confirm their conser-660 vation, consistency, and boundedness properties. It is particularly important that the $w$-level dual mass budget is satisfied (section 2.2) so that $w$-level scalars are correctly advected. Conservation, consistency, and boundedness have also been verified in full model simulations to ensure that they are maintained by both the full cascade advection and the cheap advection updates.

### 4.2 Divergence-consistency

It has been verified that the divergent velocity increments discussed in section 2.6 correctly compensate for any discrepancy be-665 tween the departure volumes computed by SLICE and the trajectory-average divergence. When these compensating increments are not included the model is found to become unstable, as expected theoretically.





### 4.3 Thermodynamics and linearized thermodynamics

The correctness and consistency of the formulation of thermodynamics in terms of internal energy potentials was verified in stand-alone code before incorporating in PTerodaC$^3$TILES. The use of switched constraints and partial increments within the quasi-Newton solver allows $q^l$ and $q^f$ to be zero but not negative, as intended. For the PTerodaC$^3$TILES implementation, the correctness of the various test case initial states, constructed to be in hydrostatic balance and thermodynamic equilibrium, and the broadly correct evolution of thermodynamic profiles and clouds provide further verification of the thermodynamics.

The correct linearization of the thermodynamics encoded in the matrix $\widetilde{\mathbf{M}}$ is critical to the success of the formulation but is fiddly and susceptible to errors. The linearization has been verified by comparing the actual changes in the left hand sides of (6)-(13) with the changes predicted by the linearization when individual thermodynamic variables are perturbed. This testing considered base states with and without liquid and frozen water to ensure that all cases were covered.

### 4.4 Solver convergence

Good solver convergence is also critical for the success of the formulation. A useful rule of thumb is that the residuals should decrease by roughly an order of magnitude per solver iteration so that only a small number of iterations is needed per time step. However, ensuring correct performance of the solver is far from trivial, as it depends on correct formulation and implementation of the linearization, derivation of the Helmholtz problem, solution of the Helmholtz problem, and backsubstitution. Therefore, the ability of the solver to correct known errors was directly verified, as follows.

A known solution for the model state at step $n+1$ is required. This may be a known steady state such as a horizontally uniform state in hydrostatic balance, or a more complex three-dimensional state obtained by taking a sufficiently large number of solver iterations. When a small perturbation is made to this state, the solver should be able to fix that perturbation almost completely in one iteration. Testing included perturbing each model state variable in turn, and considered perturbations with different spatial structures, including globally uniform, horizontally uniform vertically localized, horizontally localized vertically uniform, and localized at a single point in the domain interior or at a top or bottom boundary.

Figure 5 shows the maximum residuals in the $q$ and $u$ equations and the maximum pressure increment versus iteration number at hour 9 of the ARM case when both the dynamics and moist thermodynamics are very active. For the purpose of calculating these diagnostics the model was restarted for a single time step and 12 solver iterations were taken rather than the default 3. There is a very large drop in the maximum $q$ residual between the first and second iterations associated with the cheap transport updates (section 2.6). There is a rather small reduction in the pressure increment between the first and second iterations, probably related to constraint switching in the thermodynamics. Otherwise, all three quantities steadily decrease by nearly an order of magnitude per iteration. Note that the residuals and pressure increments calculated at the fourth iteration are a measure of the errors due to incomplete solver convergence that remain upon completion of the third iteration.

As discussed in section 2.11, when the thermodynamic state switches between absence and presence of condensate a subset of thermodynamic variables local to the switch receive only partial increments. It is important to understand the extent to which such partial increments adversely affect the solver convergence. The DYCOMS case is especially challenging in this





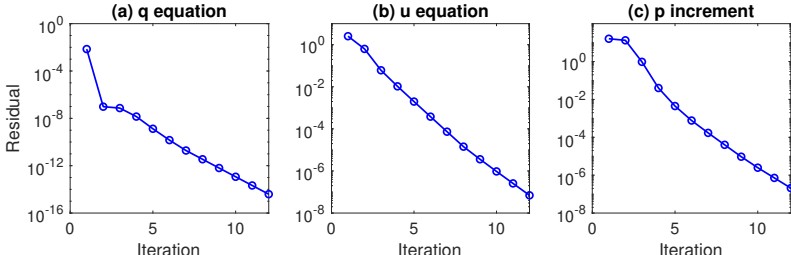

**Figure 5.** Maimum absolute value within the model domain of (a) $q$ equation residual (dimensionless) (b) $u$ equation residual $(\mathrm{ms}^{-1})$ and (c) $p'$ (Pa) versus iteration number. These diagnostics were computed for a single time step, restarting from hour 9 of the ARM case and taking 12 solver iterations.

**Table 1.** Representative number of grid points at which switches occurred at different solver iterations. The numbers were monitored over 10 time steps one hour into the DYCOMS case. The total number of model gridpoints was $128 \times 128 \times 300$.

| Iteration number | Number of switches |
|---|---|
| 1 | $\sim 60000$ |
| 2 | $\sim 600$ |
| 3 | $\sim 10$ |
| 4 | $\sim 0$ |

regard. If the cloud deck begins to break up, then holes form in the cloud (e.g., Fig. 12). If the horizontal advective Courant number is greater than 1, then a very large number of grid cells per step switch between absence and presence of liquid water. Moreover, the extremely large gradients in humidity and entropy at cloud top, which get folded into the cloud holes, make the thermodynamic state at the cloud top and cloud hole edges very sensitive to the solver advection updates, so that switching can occur at the second and subsequent solver iterations.

An initial attempt to run the DYCOMS case at $128 \times 128 \times 300$ resolution ($\Delta x = \Delta y = 25$m, $\Delta z = 5$m) with $N_\ell = 3$ solver iterations resulted in failure of the model after about 32 minutes. Diagnostics indicated that the switching had not completely settled down after 3 solver iterations. Nevertheless, the number of switches decreases by two orders of magnitude per iteration, and increasing the number of iterations to $N_\ell = 4$ is sufficient for virtually all of the switching to settle down (Table 1), allowing the DYCOMS case to run successfully. Although constraint switching can slow solver convergence, no cases have
been encountered in which constraint switching prevents solver convergence.

### 4.5 Stability

The semi-implicit semi-Lagrangian formulation of PTerodaC$^3$TILES should be stable for large acoustic, gravity-wave, and advective Courant numbers. Diagnostics optionally output by the model confirm that the model does run stably with advec-





tive Courant numbers greater than 1 and with very large acoustic Courant numbers. For example, in the ARM case presented
in section 5.2 the horizontal and vertical acoustic Courant numbers remain around 53 and 87, respectively. The horizontal
advective Courant number in the $x$-direction varies in the range 1.5 to 2, while the vertical advective Courant number peaks
at about 3. Both the gravity-wave and convective Courant numbers $\sqrt{\max(N^2)}\Delta t$ and $\sqrt{\max(-N^2)}\Delta t$, respectively, peak
at around 0.6. For the DYCOMS case presented in section 5.3 the horizontal and vertical acoustic Courant numbers remain
around 49 and 344, respectively. The horizontal advective Courant numbers in both the $x$- and $y$-directions are greater than 1,
while the vertical advective Courant number peaks at over 3. The gravity-wave Courant number is about 0.9 while the convec-
tive Courant number varies between 0.8 and 1.4.

Because the ILES formulation does not include an eddy-diffusion-based subgrid scheme, the diffusive Courant number is
not relevant for model stability.

The model does fail for time steps that are too large, for example if the time step is increased to 6s or greater in the DYCOMS
case with other parameters as in section 5.3. Apart from the use of a too small value of $N_\ell$ when there is strong constraint
switching, as discussed in section 4.4, the other factor likely to limit the model's stability is the deformational Courant number.
Specifically, the semi-Lagragian trajectory calculations will become increasingly inaccurate as the eigenvalues of $b\Delta t\nabla\boldsymbol{u}$
approach and exceed 1. Diagnostics of the components of $\Delta t\nabla\boldsymbol{u}$ for the cases presented in section 5 suggest that the model is
close to that regime in the surface layer where vertical shear is strongest.

## 4.6 Conservation

PTerodaC$^3$TILES is designed to have closed budgets for mass, water, and entropy. This property requires conservative advec-
tion of these quantities by both SLICE and the cheap transport updates (section 2.6), as well as careful formulation of surface
and interior sources (section 2.7), all taking into account the $w$-level dual mass budget (section 2.2).

Figure 6 shows time series of various budget quantities for the ARM case. Panel (a) shows that the change in total mass
(blue curve) agrees with the accumulated source of total mass, in this case due to the surface moisture flux (blue symbols),
while the dry mass is exactly conserved. Panel (b) shows that the change in total water mass (black curve) agrees with the
accumulated source of water (black symbols), even while some water condenses into liquid (red curve). Panel (d) shows that
the change in total entropy (black curve) agrees with the accumulated source of entropy (black symbols). Panel (c) confirms
that the momentum budget is not closed because the semi-Lagrangian advection of momentum is not conservative. Panel (e)
shows that the change in total energy (black curve), which averages $432\mathrm{Wm}^{-2}$, is slightly less than the accumulated total
energy source (black symbols). The difference, shown in panel (f), is due to numerical dissipation of kinetic energy and mixing
of water and entropy. This energy loss averages about $12\mathrm{Wm}^{-2}$.

Dissipation of kinetic energy and mixing of constituents and heat should result in a source of entropy that exactly compen-
sates this energy loss; however, this entropy source is currently neglected. See further discussion in section 6. Because the
current formulation assumes thermodynamic equilibrium in each grid cell, entropy sources due to departures from equilibrium
(the $\boldsymbol{J}^{\mathrm{T}}\boldsymbol{P}$ term in (4)) are zero.



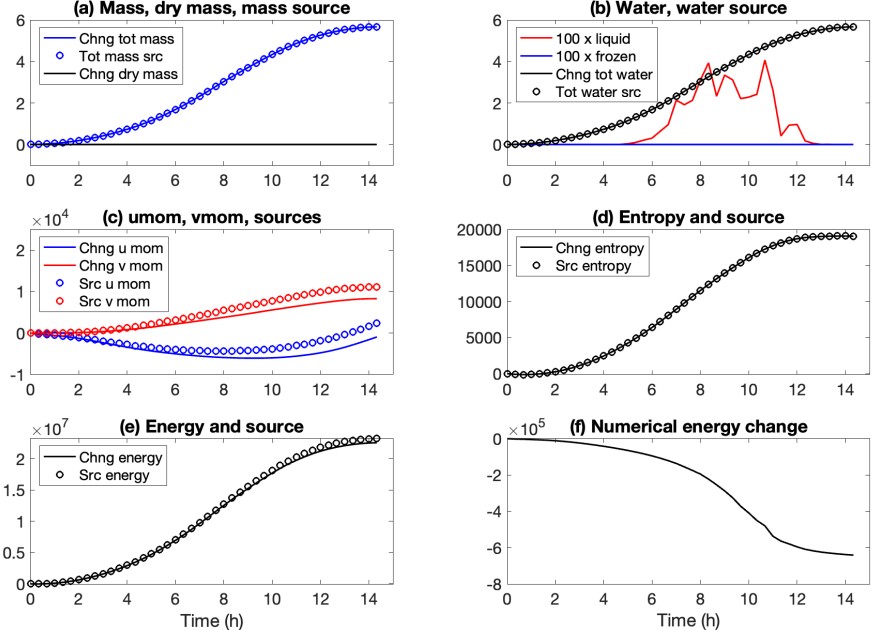

**Figure 6.** Global changes and accumulated sources versus time for key budget quantities for the ARM case. (a) Total mass and dry mass $(\text{kgm}^{-2})$. (b) Total water; $100\times$ global liquid water and frozen water $(\text{kgm}^{-2})$. (c) $u$-momentum and $v$-momentum $(\text{kgms}^{-1}\text{m}^{-2})$. (d) Entropy $(\text{JK}^{-1}\text{m}^{-2})$. (e) Energy $(\text{Jm}^{-2})$. (f) Numerical energy change, i.e., the actual energy change minus the change due to explicit sources $(\text{Jm}^{-2})$.

### 4.7 Bit-reproducibility

PTerodaC$^3$TILES produces bit-reproducible results when restarted from a checkpoint file[2]. This property is invaluable for development and testing, as well as when re-running sections of a simulation to obtain additional diagnostics. PTerodaC$^3$TILES also produces bit-identical answers whether run with or without OpenMP shared memory parallelism, verifying the correctness of the parallel implementation.

### 5 Evaluation

This section presents results from some standard LES test cases to demonstrate the performance of PTerodaC$^3$TILES version 1.0. The same spatial resolution is used as in the original intercomparison articles defining the test case specifications.

---

[2]In theory, the multigrid solver parameters automatically set by the model could change upon restart, breaking bit-reproducibilty. The author has not noticed any cases where this happens. Nevertheless, this loophole should be closed in a future model version.





Although relatively coarse by current day standards, this facilitates comparison with those published results and helps to highlight any limitations of PTerodaC$^3$TILES version 1.0 that might be less conspicuous at finer resolution.

All results shown in this section used SLICE advection with piecewise constant remapping for density and parabolic spline remapping with a limiter and the Charney-Phillips grid correction for water and entropy. Semi-Lagrangian advection of velocity components used no limiter and the `'freeslip'` option for extrapolation near the bottom boundary.

## 5.1   BOMEX

The BOMEX test case (Siebesma et al., 2003) is based on observations made during the Barbados Oceanographic and Meteorological Experiment. It simulates a scenario of shallow cumulus over the ocean in which large scale forcing, radiation, and turbulent and convective fluxes maintain a quasi-steady balance.

As in Siebesma et al. (2003), PTerodaC$^3$TILES used a $64 \times 64 \times 75$ grid with $\Delta x = \Delta y = 100$m, $\Delta z = 40$m. The time step
was $\Delta t = 10$s and the simulation was run for 6h.

Figure 7 shows time series of three key quantities: total cloud cover, liquid water path (LWP), and tubulent kinetic energy (TKE). All three time series agree broadly with Fig. 2 of Siebesma et al. (2003). After an initial spin-up during the first hour or so, the total cloud cover and LWP fluctuate but have little trend, while the TKE continues to grow slowly. The PTerodaC$^3$TILES cloud cover is slightly lower and the TKE slightly larger than the ensemble means in Siebesma et al. (2003), but within the
typical inter-model spread.

Figure 8 shows several horizontally averaged profiles from the BOMEX case. Panels (b), (c), and (d) agree well with the corresponding figures from Siebesma et al. (2003) (their figures 6, 3(d), and 4(a)). Panels (a), (e), and (f) also broadly agree with the corresponding figures from Siebesma et al. (2003) (their figures 3(c), 4(e), and 5(a)). However, an excessively strong shear layer has formed between model levels 1 and 2, consistent with the idea that the ILES approach poorly represents vertical
subgrid transports near a horizontal boundary. This excessively strong shear layer leads to noise in the lowest two to three levels in the profiles of $u$, momentum flux, and TKE.

On a closely related point, numerical experimentation revealed that the momentum budget and the boundary layer $u$ profile are strongly sensitive to the details of the semi-Lagrangian interpolation scheme for velocity components near the bottom boundary (section 2.6). For example, switching on the limiter for velocity advection or using the `'noslip⎵⎵'` bottom bound-
ary extrapolation option resulted in a large spurious numerical source of eastward momentum and a significant shift to the right of the boundary layer $u$ profile.

## 5.2   ARM

The ARM test case (Brown et al., 2002) simulates the diurnal evolution of shallow convection over land, starting from an initially stable and cloud-free boundary layer. The number, size, and depth of clouds evolve during the day, and clouds often
overshoot their level of neutral buoyancy.



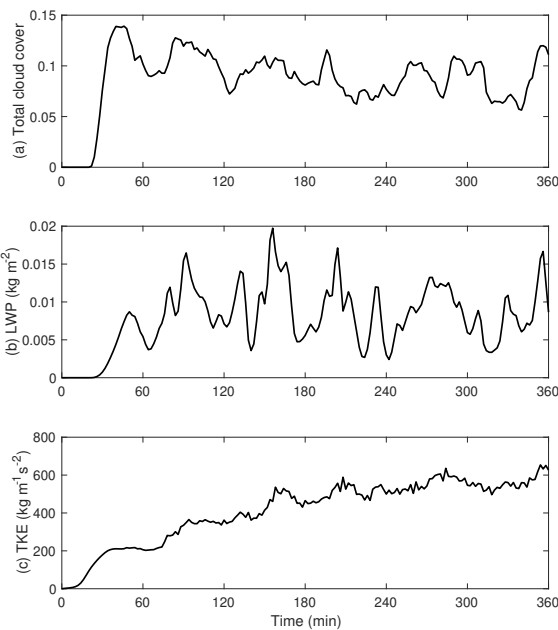

**Figure 7.** Time series of (a) Total cloud cover, (b) Liquid water path, (c) Turbulent kinetic energy for the BOMEX case.

As in Brown et al. (2002), PTerodaC$^3$TILES used a $96 \times 96 \times 110$ grid with $\Delta x = \Delta y = 66.7$m, $\Delta z = 40$m. The time step was $\Delta t = 10$s and the simulation was run for $14.5$h. Large-scale forcing terms were omitted, since they have only a small effect on the simulation (Brown et al., 2002).

Figure 9 shows a time-height plot of cloud fraction and TKE. The evolution of the cloud fraction, which peaks at a little

over 0.15 around 6 or 7h, as well as the height of cloud base and cloud top, agree well with Fig. 5 of Brown et al. (2002). The intensity of TKE in the boundary layer grows and then decays in concert with the strength of the surface heat flux, and there is a clear signature of the formation of gravity waves in and above the cloud layer.

Figure 10 shows profiles of $u$- and $w$-variances at 3h and 9h. The peak in $u$-variance at level 2 ($z = 60$m) is somewhat larger than in Brown et al. (2002) (their Fig. 6), but otherwise these profiles agree well with Brown et al. (2002), including

795 the secondary peak in $w$-variance in the cloud layer at 9h, and the small peak in $u$-variance near the boundary layer top at 3h, which disappears after clouds form (see discussion in Brown et al., 2002, section 3(b)).

Figure 11 shows snapshots of cloud top height at selected times. The behaviour is consistent with that documented for the ARM case, with many small and shallow clouds at early times, gradually growing in depth, evolving towards fewer larger, deeper clouds with smaller total cloud cover at later times. The horizontal motion of the higher cloud tops is almost exactly in

the $x$-direction (the direction of the background geostrophic wind) while near cloud base there is a small component of motion in the $y$-direction, leading to a distinct characteristic tilt to the clouds.



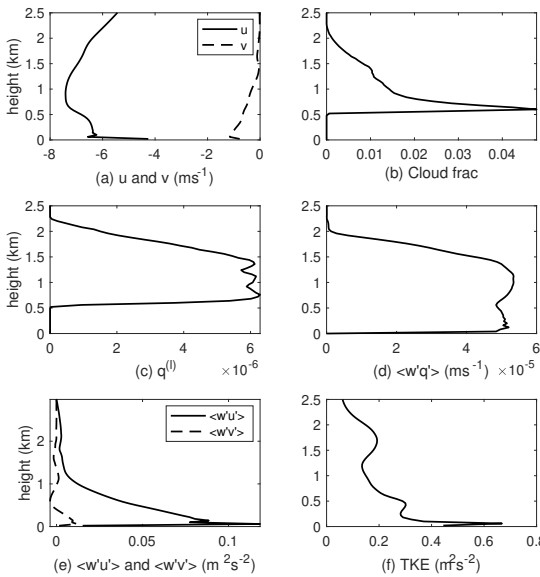

**Figure 8.** Vertical profiles of horizontally averaged quantities from the BOMEX case. (a) $u$ and $v$ $(\mathrm{ms}^{-1})$. (b) Cloud fraction (dimensionless). (c) Liquid water $q^{\mathrm{l}}$ (dimensionless). (d) Vertical eddy flux of total water $\langle w'q' \rangle$ $(\mathrm{ms}^{-1})$. (e) Vertical eddy fluxes of momentum $\langle w'u' \rangle$ and $\langle w'v' \rangle$ $(\mathrm{m}^2\mathrm{s}^{-2})$. (f) TKE $(\mathrm{m}^2\mathrm{s}^{-2})$. (a), (b) and (c) are averaged over the last hour of simulation; (d), (e) and (f) are averaged over the last three hours.

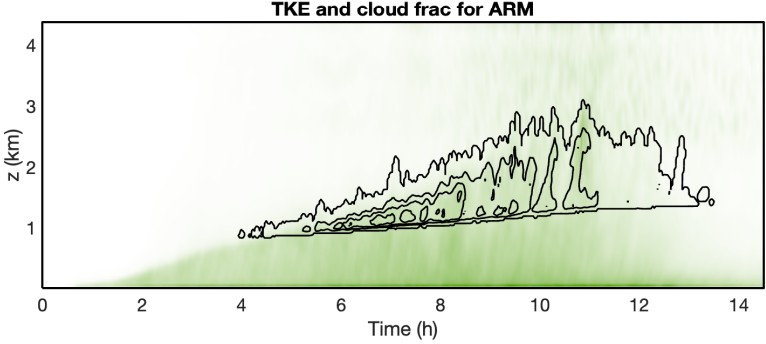

**Figure 9.** TKE (shading) and cloud fraction (contours) versus time and height for the ARM case. The contour values are 0.0001, 0.05, 0.1, and 0.15. The peak TKE value is $3.29\mathrm{m}^2\mathrm{s}^{-2}$.

## 5.3  DYCOMS

The DYCOMS test case (Stevens et al., 2005) simulates a very different boundary layer regime from BOMEX and ARM: a nocturnal stratocumulus cloud layer over the ocean. The cloud layer is capped by a very strong and sharp inversion, with $q$



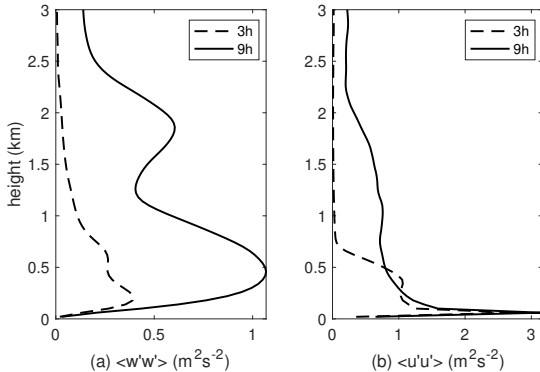

**Figure 10.** Vertical profiles of (a) $w$-variance and (b) $u$-variance for the ARM case at 3h and 9h. The profiles are also averaged in time between minus and plus 30min of the nominal diagnostic time.

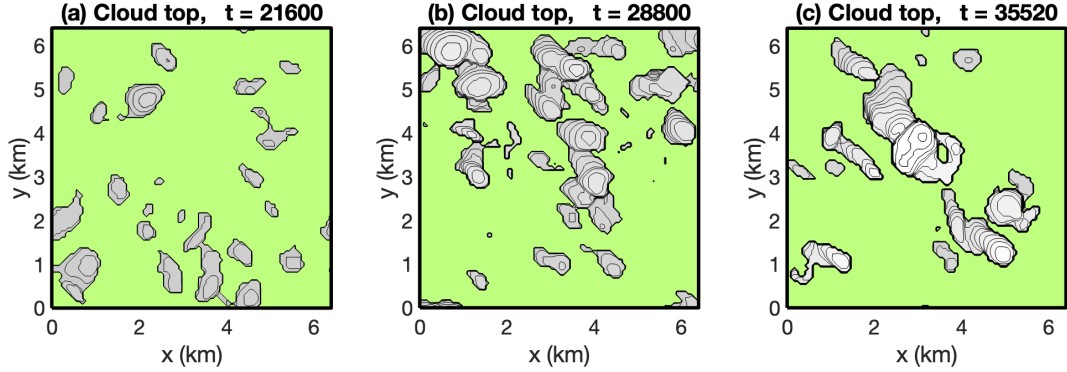

**Figure 11.** Snapshots of cloud top height (grey) for the ARM case at 6h, 8h, and 9h 56min. For this purpose, grid cells with $q^{\mathrm{l}} + q^{\mathrm{f}} > 10^{-5}$ are defined to be cloudy (though $q^{\mathrm{f}} \equiv 0$ for ARM). The green background indicates no cloud in that column, and the contour interval is 100m; refer to Fig. 9 for the lowest cloud base and highest cloud top at these times.

decreasing by a factor of 6 and a jump in potential temperature of more than 9K over 5m, the recommended vertical grid spacing. Observations suggest that the cloud cover should be maintained close to 100%. However, LES often fail to maintain the cloud cover, since excessive mixing across the inversion can lead to evaporative cooling driving cloud-free downdrafts.

The DYCOMS case is expected to be particularly testing for the PTerodaC$^3$TILES formulation, for several reasons. First, there are no physical parameters that can be adjusted to control the strength of mixing near the inversion, since the mixing is
810 entirely associated with the numerics. There is, however, some sensitivity to the choice of numerical options and parameters, as discussed below. Second, good convergence of the semi-implicit iterative solver depends on having a sufficiently good linearization. The presence of near discontinuities in the distributions of total entropy and total specific humidity mean that the linearized advection terms (41) and (42) are necessarily less accurate. The solver convergence is also affected by the switching



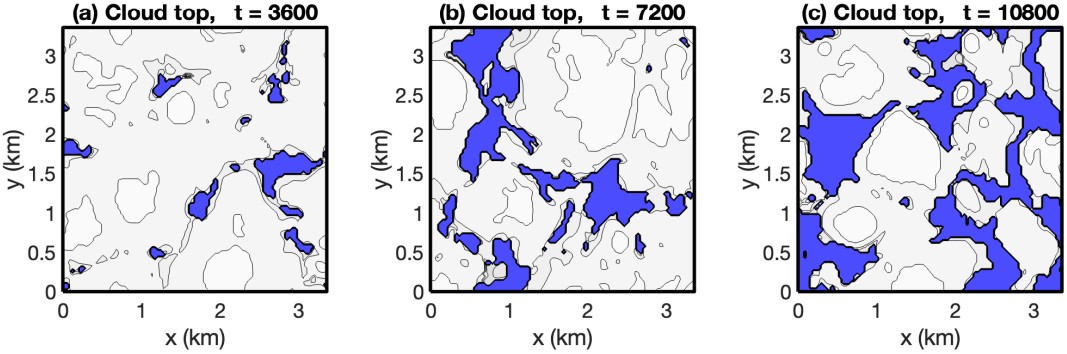

**Figure 12.** Snapshots of cloud top height (pale grey) for the DYCOMS case at 1h, 2h, and 3h. The blue background indicates no cloud in that column, and the contour interval is 50m; refer to Fig. 14 for the lowest cloud base and highest cloud top at these times.

of thermodynamic constraints (sections 2.11 and 4.4). Once cloud-free downdrafts form (Fig. 12), and are advected across
the grid with horizontal Courant numbers greater than 1, condensate appears or disappears at large numbers of gridpoints
every step. Moreover, the large gradients in entropy and specific humidity mean that transport increments resulting from small
velocity increments at one solver iteration are enough to cause constraint switching at the next solver iteration. Experience to
date suggests that more solver iterations per step are required for DYCOMS than for other test cases in order for the constraint
switching to settle down (section 4.4).

As in Stevens et al. (2005), PTerodaC$^3$TILES used a $96 \times 96 \times 300$ grid with $\Delta x = \Delta y = 35$m, $\Delta z = 5$m. The time step
was $\Delta t = 5$s and the simulation was run for 4h.

Figure 13 (compare Stevens et al., 2005, Fig. 2) shows that in the PTerodaC$^3$TILES simulation cloud cover gradually falls
to about 40% while the liquid water path falls to around $0.02$kgm$^{-2}$ (about $0.035$kgm$^{-2}$ for the ensemble mean in Stevens
et al. (2005)). The TKE settles down at about $200$kgm$^{-1}$s$^{-2}$, about half of the ensemble mean in Stevens et al. (2005) (noting
the different units used). The gradual reduction in cloud cover is also clearly illustrated in Fig. 14. The figure confirms that
turbulence grows initially both from the surface (due to the surface sensible heat flux) and from cloud top (due to cloud top
radiative cooling).

The PTerodaC$^3$TILES simulation of cloud in the DYCOMS case appears to be quite sensitive to the choice of numerical
options and parameters. For example, when the time step is reduced from 5s to 2s, the cloud breakup is slowed, leaving 70%
cloud cover after 4h. On the other hand, when parabolic spline remapping is replaced by piecewise parabolic method remapping
for the advection of entropy and water almost all cloud disappears after 4h. Further experiments and diagnostics to understand
these sensitivities would be valuable.

### 5.4 Computational cost

While noting the usual caveats that computational costs are sensitive to computing platform, compiler, output files written,
and many other factors, it is nevertheless useful to give would-be users an idea of the computational cost of PTerodaC$^3$TILES.



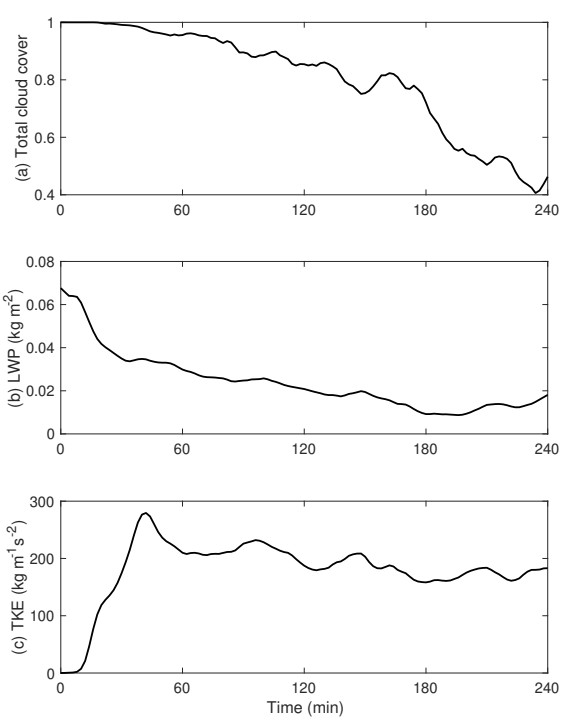

**Figure 13.** Time series of (a) total cloud cover, (b) liquid water path, (c) turbulent kinetic energy for the DYCOMS case.

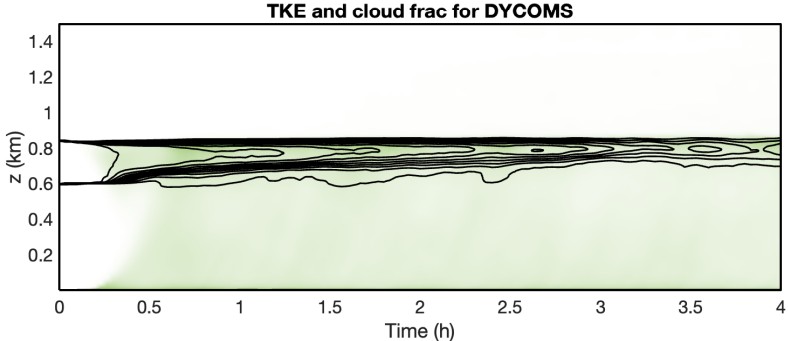

**Figure 14.** TKE (shading) and cloud fraction (contours) versus time and height for the DYCOMS case. The contour values are 0.3 to 0.9 in steps of 0.1, and 0.99. The peak TKE value is $1.16 \mathrm{m}^2 \mathrm{s}^{-2}$.

Table 2 gives the wall-clock times for the three test cases discussed in this section. The cases were run on a 2022 MacBook Pro with an 8 core M2 chip. The gfortran compiler was used with the -O3 optimization flag and OpenMP for shared memory parallelism. The parallel speed-up ranged from 574% for BOMEX to 652% for DYCOMS.





**Table 2.** Computational cost of PTerodaC³TILES for three standard test cases on a 2022 MacBook Pro laptop.

| Case | Number of grid cells | Number of time steps | $N_\ell$ | Wall-clock time (s) |
|------|---------|---------|------|---------|
| BOMEX | $64 \times 64 \times 75$ | 2160 | 3 | 903 |
| ARM | $96 \times 96 \times 110$ | 5220 | 3 | 6840 |
| DYCOMS | $96 \times 96 \times 300$ | 2880 | 4 | 12659 |

The multigrid elliptic solver and the full transport calculation dominate the cost, accounting for approximately 40% and 35% of the total cost, respectively, with some variation from case to case. Creating the thermodynamic subsystem matrix and carrying out the Gaussian elimination account for less than 15% of the total cost.

## 6 Conclusions and discussion

It has been demonstrated that a consistent treatment of moist thermodynamics, expressed in terms of internal energy potentials, can be incorporated, without excessive computational expense, in a three-dimensional computational fluid dynamics code suitable for the study of atmospheric boundary layers and shallow convection. In the current implementation the moist thermodynamics is fully incorporated within the dynamical core rather than treated as separate 'physics' source terms.

The iterative solver for the semi-implicit time integration scheme requires a linearization of the thermodynamics, with elimination of unknowns to leave a typical Helmholtz problem for the pressure increment. The moderate sparsity and fixed sparsity pattern of the Jacobian matrix of the ($w$-level) thermodynamic subsystem are exploited to reduce the cost of the elimination.

The use of internal energy potentials has several advantages over the Gibbs function approach used by Thuburn (2017b) (see section 1). However, one disadvantage of the internal energy potential approach is that it does not permit seamlessly switching to a (quasi-)Boussinesq equation of state since $e_\alpha$ terms (used throughout the algorithm) would become undefined. A nearly Boussinesq fluid could be simulated by making the sound speed extremely large. Alternatively, if a Boussinesq option is a requirement, then the above difficulty could be avoided by using specific enthalpies as the potentials in formulating the thermodynamics (Chris Eldred, personal communication, 2024).

The numerical methods used by PTerodaC³TILES are more typical of those used in global weather and climate models than traditional LES models. The semi-implicit semi-Lagrangian scheme permits time steps significantly larger than are commonly used in traditional LES models at similar resolution. Experience to date, consistent with the theoretical properties of the numerical methods, suggests that the deformational Courant number is most often the factor limiting the maximum stable time step. However, there is no simple, easily monitored stability criterion (computing the eigenvalues of $\Delta t \nabla \boldsymbol{u}$ at every grid point and every step would be expensive), so a degree of experimentation is required to find a suitable time step for each simulation. An important caveat is that stability does not imply accuracy. For example, the DYCOMS case (section 5.3) shows significant





sensitivity to the size of the time step. Further investigation is required to quantify the extent to which accuracy declines as we push the time step towards the stability limit, and which flow regimes, like DYCOMS, are especially sensitive to the time step.

For the BOMEX, ARM, and DYCOMS cases presented here, and other cases not discussed, PTerodaC$^3$TILES can produce plausible simulations of standard LES test cases, comparable to other model results in the literature. These results are encouraging for the ILES approach and, more widely, for the use of global models at sub-kilometre resolution. At the same time, these results have given some useful initial indications of the limitations of the ILES approach as well as highlighting areas in need of further investigation.

Most notably, ILES produces weak vertical fluxes of momentum and scalars near the bottom boundary. These weak fluxes lead to excessive vertical gradients, which, in turn, result in further errors. For example, in BOMEX the spuriously strong vertical shear exacerbates the conservation errors of the cubic semi-Lagrangian advection of momentum, and makes the results very sensitive to the details of the semi-Lagrangian interpolation in the lowest levels. In an ARM simulation with finer vertical resolution near the surface (not shown), the spuriously weak mixing near the surface allows a thin layer of fog to form briefly during the first hour. Some initial experimentation showed benefits in distributing the convergence of the surface momentum flux over several layers near the surface rather than just a single layer. In future work it is planned to investigate this idea further, applied to scalars as well as momentum, and to try to determine how the optimal distribution depends on the flow regime as well as numerical factors such as the grid resolution and anisotropy.

Another broad area in need of further investigation is the sensitivity to numerical methods and parameters. The sensitivity of the DYCOMS case to the time step and the advection remapping scheme has been mentioned already. Other potentially significant factors include the grid isotropy, and the use of different advection options such as the use of limiters and the Charney-Phillips grid correction (27), and the modifications to momentum interpolation near the bottom boundary.

Taking inspiration from global models that predict potential temperature, PTerodaC$^3$TILES has a closed budget for entropy rather than energy. The entropy source that should be associated with numerical mixing of scalars and dissipation of kinetic energy is neglected, resulting in a small but systematic energy loss (Fig. 6). If desired, the *global* numerical energy loss could easily be diagnosed and returned as an entropy source to close the energy budget. Initial attempts to diagnose the *local* numerical energy loss (not shown) suggest that the calculation is subtle and far from trivial, and might not even be well-defined, in large part because of the Charney-Phillips vertical grid staggering. Further work will be needed to diagnose the local numerical energy loss, if, indeed, it is possible at all.

Two closely related priorities for future work are the inclusion of a simple microphysics scheme with precipitation, and the inclusion of thermodynamic nonequilibrium effects. These developments will enhance the capabilities of PTerodaC$^3$TILES, allowing it to be applied to a wider range of cases. Equally importantly, they will test whether the thermodynamic potential approach can be applied straightforwardly and efficiently to more complex physical processes, beyond the coupling of the equation of state to the dynamical core.





*Code availability.* The exact version of the model code PTerodaC³TILES 1.0 used to produce the results in this paper, an example namelist file, example plotting routines, and the User Manual, are available from Zenodo: https://doi.org/10.5281/zenodo.13899067. The code is made available under the MIT licence.

The namelist files needed to create initial data for the BOMEX, ARM, and DYCOMS cases and to reproduce the simulations presented here, along with the plotting routines used to produce figures 5 to 14, are provided in the supplementary materials.

## Appendix A

### A1   Stretched vertical grid

When a stretched vertical grid is chosen, the height of the model level with index $k$ is given by

$$z_k = a_{(z)} \log\left(1 + d_{(z)} b_{(z)}^k\right) + c_{(z)}, \tag{A1}$$

where $k$ is an integer for $p$-levels and an integer plus $1/2$ for $w$-levels. The parameters in (A1) are set so that the grid spacing approaches a stretching factor $b_{(z)}$ per level for small $k$ and a uniform grid spacing $u_{(z)}$ for large $k$, with $b_{(z)}$ and $u_{(z)}$ specified by the user when creating initial data. In terms of $N_z$ the number of $p$-levels and $D_z = z_{N_z+1/2} - z_{1/2}$ the domain depth with $z_{1/2} = 0$ the height of the bottom boundary, the parameters are given by

$$a_{(z)} = u_{(z)} / \log b_{(z)}, \tag{A2}$$

$$d_{(z)} = \frac{1 - \exp(D_z/a_{(z)})}{b_{(z)}^{1/2}\left(b_{(z)}^{N_z} - \exp(D_z/a_{(z)})\right)}, \tag{A3}$$

$$c_{(z)} = D_z - a_{(z)} \log\left(1 + d_{(z)} b_{(z)}^{N_z+1/2}\right). \tag{A4}$$

Importantly, (A1) has the virtue of being invertible, allowing the cell index $k$ to be determined for any height $z$ without the
need for expensive searching in the semi-Lagrangian departure-point calculations.

### A2   Expressions for internal energy and related quantities

The expressions used for specific internal energy of dry air, water vapour, liquid water, and frozen water are

$$e^{\mathrm{d}}(\alpha^{\mathrm{d}}, \eta^{\mathrm{d}}) = C_v^{\mathrm{d}} T_0 \exp\left(\frac{\eta^{\mathrm{d}} - C_p^{\mathrm{d}}}{C_v^{\mathrm{d}}}\right)\left(\frac{\alpha_0^{\mathrm{d}}}{\alpha^{\mathrm{d}}}\right)^{R^{\mathrm{d}}/C_v^{\mathrm{d}}}, \tag{A5}$$

$$e^{\mathrm{v}}(\alpha^{\mathrm{v}}, \eta^{\mathrm{v}}) = C_v^{\mathrm{v}} T_0 \exp\left(\frac{\eta^{\mathrm{v}} - C_p^{\mathrm{v}}}{C_v^{\mathrm{v}}}\right)\left(\frac{\alpha_0^{\mathrm{v}}}{\alpha^{\mathrm{v}}}\right)^{R^{\mathrm{v}}/C_v^{\mathrm{v}}} + L_{00}^{\mathrm{s}}, \tag{A6}$$





$$e^{\mathrm{l}}(\eta^{\mathrm{l}}) = C^{\mathrm{l}} T_0 \exp\left(\frac{\eta^{\mathrm{l}} - C^{\mathrm{l}}}{C^{\mathrm{l}}} + \frac{L_{00}^{\mathrm{v}} - \alpha^{\mathrm{l}} p_0^{\mathrm{sat}}}{C^{\mathrm{l}} T_0}\right) + L_{00}^{\mathrm{f}},$$

(A7)

$$e^{\mathrm{f}}(\eta^{\mathrm{f}}) = C^{\mathrm{f}} T_0 \exp\left(\frac{\eta^{\mathrm{f}} - C^{\mathrm{f}}}{C^{\mathrm{f}}} + \frac{L_{00}^{\mathrm{s}} - \alpha^{\mathrm{f}} p_0^{\mathrm{sat}}}{C^{\mathrm{f}} T_0}\right).$$

(A8)

These expressions differ from those used by Bowen and Thuburn (2022a) in that the constant $L_{00}^{\mathrm{s}}/T_0$ has been subtracted from the specific entropy of all water phases.

For $i = \mathrm{d}$, v, l, f, the species temperature is given by

$$T^i = e_\eta^i.$$

(A9)

For $i = \mathrm{d}$, v, the species partial pressure is given by

930 $p^i = -e_\alpha^i.$

(A10)

The Gibbs function for water vapour is given by

$$g^{\mathrm{v}} = e^{\mathrm{v}} + \alpha^{\mathrm{v}} p^{\mathrm{v}} - \eta^{\mathrm{v}} T^{\mathrm{v}},$$

(A11)

while for $i = \mathrm{l}$, f,

$$g^i = e^i + \alpha^i p - \eta^i T^i,$$

(A12)

with $p = p^{\mathrm{d}} + p^{\mathrm{v}}$. (The Gibbs function for dry air is not needed.)

### A3 Parameters for multigrid scheme

The tuning of parameters in numerical solvers for elliptic problems is often empirical and case-dependent. Here, both the Helmholtz problem and the properties of the multigrid solver are well understood, which allows some key parameters in the multigrid scheme to be set automatically.

Let $C_{\mathrm{ac}}^x = c\Delta t/\Delta x$, $C_{\mathrm{ac}}^y = c\Delta t/\Delta y$ be the horizontal acoustic wave Courant numbers in the $x$- and $y$-directions, respectively, appropriate to whichever grid in the hierarchy is under consideration. Let $C_{\mathrm{ac}} = \max(C_{\mathrm{ac}}^x, C_{\mathrm{ac}}^y)$. In the following calculations the temporal off-centring parameter $a$ is effectively set to 1.

Each smoother iteration uses a direct solve in the vertical direction; thus, we must consider the action of the smoother on errors of different horizontal scales. The error amplification factor for a uniform error for one Jacobi smoother iteration is

945 $A_\varepsilon = 1 - \dfrac{\mu}{1 + 2\left((C_{\mathrm{ac}}^x)^2 + (C_{\mathrm{ac}}^y)^2\right)},$

(A13)

where $\mu$ is the under-relaxation parameter, while the error amplification factor for a checkerboard pattern error is

$$A_\varepsilon = 1 - \mu - \frac{2\mu\left((C_{\mathrm{ac}}^x)^2 + (C_{\mathrm{ac}}^y)^2\right)}{1 + 2\left((C_{\mathrm{ac}}^x)^2 + (C_{\mathrm{ac}}^y)^2\right)}.$$

(A14)





1. **Under-relaxation parameter**. The under-relaxation parameter is set to $\mu = 0.8$. This provides an appropriate compromise between the damping of large-scale errors on the coarsest grid, where $C_{ac} \approx 1$ implies $A_\varepsilon \approx 0.85$, and the damping of grid-scale errors on the finest grid, where $C_{ac} \gg 1$ implies $A_\varepsilon \approx 0.6$.

2. **Depth of V-cycles**. As the horizontal grid is coarsened the horizontal acoustic wave Courant number $C_{ac}$ decreases, the horizontal part of the Helmholtz problem becomes more diagonally dominant, and the smoother iterations damp the error more and more quickly. Once the horizontal acoustic Courant number reaches about 1, there is little to be gained by coarsening the grid further. Thus, the desired V-cycle depth is set to the smallest depth needed to reduce the horizontal acoustic Courant number below 1. The maximum possible V-cycle depth permitted by the number of grid points in the $x$- and $y$-directions is also computed. The actual V-cycle depth is set equal to the minimum of the desired depth and the maximum permitted depth. If the maximum permitted depth is smaller than the desired depth then a warning message printed and the scheme attempts to compensate by increasing the number of coarsest grid smoother iterations.

3. **Number of coarsest grid smoother iterations**. The rate at which large-scale errors are damped is determined by the damping of uniform errors on the coarsest grid. Using (A13) to estimate the large-scale error damping rate, the number of smoother iterations on the coarsest grid is chosen so that a uniform error is damped by a factor $1/10$ on each V-cycle.

4. **Number of V-cycles**. To ensure mass conservation, the backsubstitution computes density increments from the divergence of mass flux increments, with velocity increments computed from the gradient of pressure increments. As a consequence of the gradient and divergence operations, any grid-scale errors in the pressure increments due to imperfect convergence of the multigrid solver get amplified by a factor equal to the (fine-grid) horizontal acoustic Courant number squared. Thus, to ensure good convergence of the quasi-Newton solver, any grid-scale errors in the pressure increments computed by the multigrid solver must be much smaller, by a factor an order of magnitude smaller than $1/C_{ac}^2$, than the pressure increments themselves. Using the estimate from (A14) that $|A_\varepsilon| \approx |1 - 2\mu|$ for large $C_{ac}$, we can estimate the total number of fine-grid smoother iterations required, and hence the number of V-cycles.

## A4 Alternative formulation of Monin-Obukhov theory for surface momentum flux

Monin-Obukhov similarity theory gives the flow speed $U(z)$ at height $z$ in terms of the friction velocity $U_*$ (e.g. Stull, 1988):

$$U(z) = \frac{U_*}{\kappa} \left\{ \log(z/z_0) + \Psi(\zeta, \zeta_0) \right\}, \tag{A15}$$

where $\kappa$ is the von Kármán constant, $\zeta = z/L$ is a non-dimensional height; $\zeta_0 = z_0/L$ where $z_0$ is the surface roughness length, $L = -U_*^3/F^b$ is the Obukhov length with $F^b$ the surface buoyancy flux, and $\Psi$ is an integrated stability function, discussed below. As written, (A15) must be solved iteratively to obtain $U_*$, given $U(z)$ at some $z$. Moreover, in the stable case ($F^b < 0$) with light winds Monin-Obukhov similarity theory is valid only for sufficiently small $z$ and may be outside its range of validity at the height of the lowest model level; in that case, with commonly used expressions for $\Psi$, the solution for $U_*$ might not be unique or might not exist (e.g., Fig. A1).





Here we focus on the case in which $F^b$ is known and the task is to determine $U_*$. The case in which $F^b$ is also to be

determined is more complicated (Bull and Derbyshire, 1990).

Since the expressions used for stability functions are typically derived by curve fitting to observations or simulations, it seems reasonable to attempt to provide an inverse to (A15) with aid of some curve fitting. The fitting can be be done in such a way as to guarantee the existence of a unique solution for $U_*$ satisfying the reasonable requirement $U_* \to 0$ as $U(z) \to 0$.

It is convenient to define $r = z_0/z = \zeta_0/\zeta$ and to introduce a non-dimensional inverse flow speed

$$\widehat{s} = \frac{(-F^b z)^{1/3}}{\kappa(U(z_1) + \varepsilon)}, \tag{A16}$$

where $\varepsilon$ is a small safety parameter to prevent infinite $\widehat{s}$ as $U(z_1) \to 0$. Then (A15) becomes

$$\widehat{s} = \frac{\zeta^{1/3}}{\{-\log r + \Psi(\zeta, r\zeta)\}}. \tag{A17}$$

To parameterize the surface momentum flux we need to be able to compute $\zeta = \zeta(\widehat{s}, r)$; $\zeta$ is a function of the two parameters $\widehat{s}$ and $r$.

To construct a functional fit for $\zeta(\widehat{s}, r)$, proceed as follows. For both the stable case and the unstable case, carry out an asymptotic expansion of (A17) for the limits of small $|\widehat{s}|$ and large $|\widehat{s}|$ and rearrange to obtain the limiting expressions for $\zeta(\widehat{s}, r)$. Then seek a simple expression that agrees with the asymptotic expressions in the two limits.

For the unstable case we begin with the integrated stability function given by Benoit (1977)

$$\Psi(\zeta, r\zeta) = \log\left\{\frac{(x_0^2 + 1)(x_0 + 1)^2}{(x^2 + 1)(x + 1)^2}\right\} + 2\left(\arctan(x) - \arctan(x_0)\right), \tag{A18}$$

where

$$x = (1 - 15\zeta)^{1/4}, \qquad x_0 = (1 - 15r\zeta)^{1/4}. \tag{A19}$$

We find

$$\zeta^{1/3} \sim \widehat{s}\left\{-\log r - \frac{15}{4}(1 - r)(\log r)^3 \widehat{s}^3\right\} \quad \text{as } \widehat{s} \to 0, \tag{A20}$$

and

$$\zeta^{1/3} \sim \widehat{s}\left\{-\frac{1}{15}\left(4(r^{-1/4} - 1)\right)^4 \widehat{s}^{-3}\right\}^{1/7} \quad \text{as } \widehat{s} \to \infty. \tag{A21}$$

These asymptotic limits are captured by the functional fit

$$\zeta^{1/3} = \left(\tilde{a} + \tilde{b}\left(\tilde{c} + (-\widehat{s})^{\tilde{p}}\right)^{\tilde{q}}\right)^{\tilde{r}}, \tag{A22}$$



with the parameters given by

$$\tilde{p} = 3; \qquad\qquad \tilde{q} = 2/7; \qquad\qquad \tilde{r} = 3/(7\tilde{p}\tilde{q}); \tag{A23}$$

$$\tilde{A} = -\log r; \qquad\qquad \tilde{B} = \frac{15}{4}(1-r)(\log r)^3; \qquad\qquad \tilde{C} = \left\{ \frac{1}{15} \left( 4(r^{-1/4} - 1) \right)^4 \right\}^{1/7}; \tag{A24}$$

$$\tilde{a} = \tilde{A}^{1/\tilde{r}} - \tilde{b}\tilde{c}^{\tilde{q}}; \qquad\qquad \tilde{b} = \tilde{C}^{1/\tilde{r}}; \qquad\qquad \tilde{c} = \left( \frac{\tilde{B}\tilde{A}^{(1-\tilde{r})/\tilde{r}}}{\tilde{b}\tilde{q}\tilde{r}} \right)^{1/(\tilde{q}-1)}. \tag{A25}$$

For the stable case we begin with a modification of the integrated stability function given by Cheng and Brutsaert (2005)

$$\Psi(\zeta, r\zeta) = a \log \left( \frac{\zeta + (c^b + \zeta^b)^{1/b}}{\zeta_0 + (c^b + \zeta_0^b)^{1/b}} \right), \tag{A26}$$

with $a = 3$, $b = 2.5$, $c = 0.5$. (The original Cheng and Brutsaert (2005) scheme corresponds to $a = 6.1$, $b = 2.5$, $c = 1$. Figure A1 shows the effect of this modification.) We find

$$\zeta^{1/3} \sim \widehat{s} \left\{ -\log r + \frac{a}{c}(1-r)(-\log r)^3 \widehat{s}^3 \right\} \quad \text{as } \widehat{s} \to 0, \tag{A27}$$

and

$$\zeta^{1/3} \sim \widehat{s} \{ -(1+a)\log r \} \quad \text{as } \widehat{s} \to \infty. \tag{A28}$$

These asymptotic limits are captured by the functional fit

$$\zeta^{1/3} = \breve{c} + \left( \breve{a} + \breve{b}\widehat{s} \right)^{\breve{p}} \Big)^{\breve{q}}, \tag{A29}$$

with the parameters given by

$$\breve{p} = 3; \qquad\qquad \breve{q} = -3; \tag{A30}$$

$$\breve{A} = -\log r; \qquad\qquad \breve{B} = \frac{a}{c}(1-r)(-\log r)^3; \qquad\qquad \breve{C} = -(1+a)\log r; \tag{A31}$$

$$\breve{a} = (\breve{A} - \breve{c})^{1/\breve{q}}; \qquad\qquad \breve{b} = \breve{B}\breve{a}^{1/\breve{q}}/q; \qquad\qquad \breve{c} = \breve{C}. \tag{A32}$$

*Author contributions.* Apart from the implementation of the dry CBL and NEUTRAL cases noted below, all work presented was carried out by the author.

*Competing interests.* The author declares that he has no competing interests.

*Acknowledgements.* The dry CBL and NEUTRAL cases were implemented by Georgios Efstathiou and Yuhang Tong, respectively.

For the purpose of open access, the author has applied a Creative Commons Attribution (CC BY) licence to any Author Accepted Manuscript version arising from this submission.



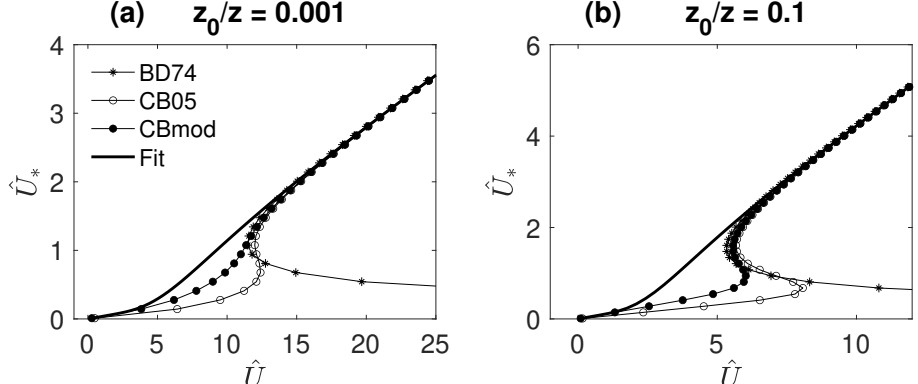

**Figure A1.** Dimensionless friction velocity $\widehat{U}_* = \zeta^{1/3}$ versus dimensionless wind speed $\widehat{U} = 1/\widehat{s}$ at height $z$ in the stable regime given by Monin-Obukhov theory with the Businger-Dyer stability function (Dyer, 1974, asterisks), with the Cheng and Brutsaert (2005) stability function (open circles), with the modified Cheng and Brutsaert stability function (A26) (filled circles), and the fit (A29) (thick curve), (a) for $r = z_0/z = 0.001$; (b) for $r = z_0/z = 0.1$. Note that with the Businger-Dyer stability function there may be zero or two solutions for $\widehat{U}_*$, while with both the original and modified Cheng and Brutsaert stability functions there may be multiple solutions for $\widehat{U}_*$.

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
