# Peer review of "Potential based Thermodynamics with Consistent Conservative Cascade Transport for Implicit Large Eddy Simulation: PTerodaC3TILES version 1.0"

_Geoscientific Model Development, 2024_

## Referee Comment (RC1)

**Review**

This paper presents a novel LES code for simulating atmospheric boundary layers and convection, focusing on three main points:

- **Thermodynamic Potentials Approach:** The code achieves thermodynamic consistency by deriving all relevant quantities from thermodynamic potentials, such as internal energy. This approach, while intricate, provides a robust framework for moist thermodynamics.

- **Semi-Implicit Semi-Lagrangian Numerics:** Departing from traditional LES models, the code employs numerical methods more commonly used in global models. This allows for larger time steps and more efficient simulations, particularly for sub-kilometer resolution cases.

- **Implicit LES:** Instead of explicit subgrid turbulence models, the code leverages dissipation from its numerical methods to represent small-scale effects. While effective, this approach exposes certain limitations near surface boundaries.

The paper also emphasizes the accessibility of the code for both research and practical applications. Features such as predefined test cases, built-in diagnostics, minimal setup requirements, and compatibility with modest computational resources enhance its usability.

Results from standard test cases demonstrate competitive performance compared to traditional LES models. However, areas for improvement are identified, particularly regarding near-surface behavior and sensitivity to numerical configurations.

The paper is exceptionally well-written and highly comprehensible, providing nearly all the details required for reproducibility.

**Comment**

The code in its current state cannot be used to run highly accurate simulations, as it is parallelized only with OpenMP shared memory parallel capability. Parallelization with MPI or, even better, MPI:GPU could significantly enhance the usability and scalability of the solver. I have tested the solver on my machine and obtained the same results as those displayed in the paper. I am therefore confident that this solver, along with all the explanations provided, will be highly beneficial for the community.

**Questions**

**General comparisons**

- Adding a small subsection that highlights a comparison with existing literature could further contextualize the code's performance and its advantages or limitations relative to established models.

**Stability**

- The author claims stability for large acoustic, gravity-wave, and advective Courant numbers, with examples provided in test cases (e.g., ARM and DY-COMS).

  - How generalizable are these stability claims? Can similar stability be achieved for cases with stronger turbulence or more complex boundary conditions, such as heterogeneous surfaces or strong temperature gradients?

  - Are there scenarios where the acoustic or advective Courant numbers exceed the reported values, leading to instability? Could the author clarify the specific limits for stability compared to other methods?

  - While the model reportedly handles advective Courant numbers $> 1$, what measures ensure numerical accuracy at these high values? For instance, how do errors in interpolation or advection affect the resolved structures?

**Timestep limitations**

- The author notes model failure when time steps are increased beyond 6 seconds in the DYCOMS case.

  - Is this limitation consistent across all cases, or does it vary depending on physical parameters (e.g., turbulence intensity, stratification, or domain resolution)?

  - Was a systematic sensitivity analysis performed to determine the optimal timestep for different types of flows? How does timestep selection affect both stability and computational cost?

**Semi-Lagrangian accuracy**

- The accuracy of semi-Lagrangian trajectory calculations decreases as eigenvalues of $\Delta t \nabla u$ approach/exceed 1.

  - Could the author quantifies the impact of these inaccuracies on key diagnostics, such as turbulent kinetic energy or scalar variance, particularly in regions where vertical shear is strongest?

**Comparisons with explicit solvers**

- The semi-Lagrangian formulation is noted for enabling large timesteps compared to explicit schemes.

  - Could the author provide direct comparisons (e.g., runtime, accuracy, or computational cost) between their method and traditional LES solvers under similar setups?

- How does the semi-implicit scheme affect the resolution of fast dynamical processes, such as wave breaking or sharp density gradients, compared to explicit solvers?

**Deformational Courant number**

- The deformational Courant number is mentioned as a potential limiting factor for stability, especially in regions with strong vertical shear.

    - How frequently does the model approach the instability threshold in practical simulations, such as those with strong surface shear or boundary-layer phenomena?
    - Are there any mitigation strategies (e.g., adaptive timestep control or trajectory smoothing) to manage cases where the deformational Courant number approaches or exceeds 1?

---

## Author Response (AR1)

**GMD-2024-153**

Final author reply to editor

J. Thuburn 4 March, 2025

Reviewer comments are in black, author responses and other comments in blue, and changes to the manuscript in red.

**Reviewer 1**

This paper presents a novel LES code for simulating atmospheric boundary layers and convection, focusing on three main points:

• Thermodynamic Potentials Approach: The code achieves thermodynamic consistency by deriving all relevant quantities from thermodynamic potentials, such as internal energy. This approach, while intricate, provides a robust framework for moist thermodynamics.

• Semi-Implicit Semi-Lagrangian Numerics: Departing from traditional LES models, the code employs numerical methods more commonly used in global models. This allows for larger time steps and more efficient simulations, particularly for sub-kilometer resolution cases.

• Implicit LES: Instead of explicit subgrid turbulence models, the code leverages dissipation from its numerical methods to represent small-scale effects. While effective, this approach exposes certain limitations near surface boundaries.

The paper also emphasizes the accessibility of the code for both research and practical applications. Features such as predefined test cases, built-in diagnostics, minimal setup requirements, and compatibility with modest computational resources enhance its usability.

Results from standard test cases demonstrate competitive performance compared to traditional LES models. However, areas for improvement are identified, particularly regarding near-surface behavior and sensitivity to numerical configurations.

The paper is exceptionally well-written and highly comprehensible, providing nearly all the details required for reproducibility.

**Thank you for the positive comments.**

**Comment**

The code in its current state cannot be used to run highly accurate simulations, as it is parallelized only with OpenMP shared memory parallel capability. Parallelization with MPI or, even better, MPI:GPU could significantly enhance the usability and scalability of the solver. I have tested the solver on my machine and obtained the same results as those displayed in the paper. I am therefore confident that this solver, along with all the explanations provided, will be highly beneficial for the community.

It was a conscious decision at the design stage to stick with OpenMP parallel in order to complete the initial development in a timely manner. An MPI extension to allow distributed memory parallelism for much higher resolution would be attractive, but would require significant code changes. Moreover, it is not clear that the cascade advection scheme would work well with MPI, so an algorithmic redesign might also be needed. Nevertheless, if the code attracts users and there is demand for greater parallelism, then this decision could be revisited in the future. An extension to use GPUs for shared memory parallelism, for example using OpenACC, would also be attractive and could perhaps be retrofitted relatively painlessly; this is something I would like to consider in the relatively near future.

Thank you for taking the trouble to install and run the code; it is good to know that it works and you were able to obtain the same results.

**Questions**

Thank you for a thought-provoking set of questions! Similar questions have been on my mind, and I wish I could give more complete answers. My student Yuhang Tong is currently working on addressing some of them. Several of the questions (Stability, Timestep limitations, Deformational Courant number) are very closely related to each other, so my responses will be similar.

**General comparisons**

• Adding a small subsection that highlights a comparison with existing literature could further contextualize the code's performance and its advantages or limitations relative to established models.

The scientific performance of PTerodaC3TILES is discussed in section 5 for the BOMEX, ARM, and DYCOMS cases. For each case, key diagnostics are plotted in the same format as the in the original intercomparison papers to facilitate comparison, and the text discusses similarities and differences compared to those papers. Limitations of the current version of PTerodaC3TILES and areas needing further investigation are highlighted in section 5 and again in the Discussion and conclusions (section 6).

The computational performance of PTerodaC3TILES is harder to compare directly with that of established LES models, since a clean comparison must be done on the same platform, and most LES codes have multiple code dependencies, requiring a non-trivial amount of work to carry out such a comparison from scratch. For this reason, the submitted paper is careful not to make direct comparisons of total cost but just notes some typical costs for PTerodaC3TILES and notes that it can take timesteps significantly larger than typical established LES codes, since published timesteps are available for standard cases at the same spatial resolutions (e.g., Stevens et al., 2005).

**Stability**

• The author claims stability for large acoustic, gravity-wave, and advective Courant numbers, with examples provided in test cases (e.g., ARM and DYCOMS).

– How generalizable are these stability claims? Can similar stability be achieved for cases with stronger turbulence or more complex boundary conditions, such as heterogeneous surfaces or strong temperature gradients?

- Are there scenarios where the acoustic or advective Courant numbers exceed the reported values, leading to instability? Could the author clarify the specific limits for stability compared to other methods?

– While the model reportedly handles advective Courant numbers > 1, what measures ensure numerical accuracy at these high values? For instance, how do errors in interpolation or advection affect the resolved structures?

The stability properties of semi-implicit, semi-Lagrangian schemes are well understood theoretically; large acoustic, gravity-wave, or advective Courant numbers, on their own, should not cause instability. However, in practical applications other factors that limit the stability (see below) also increase with  $\Delta t$ . Thus (a) it is not easy to demonstrate stability for very large acoustic, gravity-wave, or advective Courant numbers since other factors that limit stability come into play, and (b) in practice one is not likely to need Courant numbers much bigger than those seen in sections 4 and 5 of the manuscript. Nevertheless, just to emphasise the point, figures R1 and R2 show results from a rather artificial test case contrived to permit even larger acoustic, gravity-wave, and advective Courant numbers. The basic state is at rest in hydrostatic balance, and there is an extremely sharp increase in potential temperature of 20K over 3m at altitude 4.5km, with uniform potential temperature 300K below and 320K above. Onto this basic state a warm bubble of amplitude 1K is superposed. The grid spacing is 100m in the horizontal (64x64 cells) and 3m in the vertical (2000 levels). The time step was 10s. The vertical acoustic Courant number is close to 1200, and the gravity wave Courant number is greater than 3. The advective Courant number increases as the bubble accelerates upwards, peaking at over 27 before decreasing as the bubble approaches the strong stable layer and slows. Shear at the edge of the bubble increases throughout the run, and this is reflected as an increase in the deformational Courant, and, in fact, the run blows up shortly after t=600s, as expected given the large deformational Courant number at that time.

Figure R1. Entropy (colours) and vertical velocity (contours, N + 1/2 m/s for integer N, negative values dashed) for a buoyant bubble case with extremely fine vertical resolution. Left: t = 0s; right t = 600s.

Figure R2. Time series of stability-related diagnostics for the bubble case. Top left: component acoustic Courant numbers. Top right: gravity wave and convective Courant numbers. Middle left: component advective Courant numbers. Middle right, bottom left, bottom right components of the velocity gradient matrix multiplied by timestep. Colours of curves are indicated in parentheses: blue (b), red (r), green (g), black (k).

At this point, without further study, I cannot make a definitive statement about exactly what limits the stability of the model, and it probably varies from case to case. Nevertheless, based on the theoretical properties of the methods and experience to date, there are a small number of likely candidates.

(i) When the deformational Courant number approaches 1, departure points become less accurate. This might lead to badly distorted departure cells such that the simple small-amplitude advecting velocity correction cannot restore divergence-consistency (section 2.6); or there might be other feedbacks via advection that amplify errors.

(ii) If  $a^2N^2\Delta t^2$  approaches -1 then the factor  $1 + a^2N^2\Delta t^2$  in the denominator of equations (70) and (71) approaches zero and those terms blow up. This is a known limitation of the semi-implicit treatment of gravity waves when the stratification becomes unstable (e.g. Davies et al., 2005, Quart. J. Roy. Meteorol. Soc., 131, 1759-1782); it can be mitigated, for example, by bounding the value of  $N^2$  that appears in those coefficients.

(iii) Any factors that inhibit convergence of the quasi-Newton solver can mean that the theoretical stability of a semi-implicit, semi-Lagrangian scheme is not attained. This could occur, for example, (a) if the change in state over one time step is too large, hence too nonlinear, to be captured by the quasi-Newton linearization, (b) if terms omitted from the linearization, such as horizontal components of  $\mathbf{u} \cdot \nabla \eta$ , become important, or (c) if insufficient iterations are taken for constraint switching to settle down (as discussed in section 4.4).

As the reviewer suggests, stronger turbulence or more complex boundary conditions, etc., will indeed challenge the model stability more, but through these kinds of effects rather than simply

through making the Courant numbers larger.

The discussion in section 4.5 has been extended to include these points.

In contrast to implicit Eulerian advection schemes, which do tend to become more diffusive at Courant numbers > 1, semi-Lagrangian advection schemes generally retain their accuracy by moving their interpolation or remapping stencil to include the appropriate departure point or cell. (This is one of the reasons for their popularity.) Indeed, it has been argued that, provided the departure points are computed sufficiently accurately, semi-Lagrangian schemes can become MORE accurate at larger timesteps because they take fewer steps and so make fewer interpolations. Figure R3 shows another contrived experiment to illustrate the point. The setup is for the NEUTRAL boundary layer case, but with surface friction, other forcings, and initial perturbations switched off so that the flow remains steady and laminar (u, v) = (10 m/s, -5 m/s),independent of z. The grid resolution is isotropic 50m with  $128 \times 128 \times 64$  cells. The figure shows x-y slices through a passive tracer initialized with a 'pyramid' cross section. After 1280s the tracer should return to its initial position, having circumnavigated the domain twice in the xdirection and once in the y-direction. With a time step of 3.2s (component advective Courant numbers 0.64 and 0.32 in the x- and y-directions) the solution is quite accurate but there is a slight erosion of the maximum. Increasing the time step to 25.6s (Courant numbers 5.12 and 2.56) and then to 128s (Courant numbers 25.6 and 12.8) shows no loss of accuracy and, indeed, a slight improvement in capturing of the maximum.

Figure R3. Tracer distributions in the large advective Courant number test. Contour values are 0.1, 0.3, 0.5, 0.7, 0.9. Top left: initial distribution. Top right: final distribution with timestep 3.2 s. Bottom left: final distribution with timestep 25.6 s. Bottom right: final distribution with timestep 128 s.

**Timestep limitations**

• The author notes model failure when time steps are increased beyond 6 seconds in the DY-COMS case.

- Is this limitation consistent across all cases, or does it vary depending on physical parameters (e.g., turbulence intensity, stratification, or domain resolution)?

– Was a systematic sensitivity analysis performed to determine the optimal timestep for different types of flows? How does timestep selection affect both stability and computational cost?

All of the cases presented will become unstable if the timestep is set too large. However, as the reviewer suggests, the actual limit will depends on both the flow characteristics and the model resolution and other parameters. Also, it is not clear that the exact mechanism limiting the stability will be the same in all cases; see the discussion under the 'Stability' question.

A systematic sensitivity analysis has not yet been performed, but I agree that it would be good to do one. It would be a significant study in itself to answer the question properly, and I hope my student will do some of that work. It is tempting for modellers to push the time step right up to the stability limit to get faster throughput, even though it is not always clear whether accuracy is lost. (Numerical weather and climate prediction often do this, despite studies showing significant timestep sensitivity, usually associated with coupling to physical parameterisations).

Choosing a timestep that is too large always leads to instability, as noted above. Experience to date is that the onset of instability is sudden, with little warning, rather than a gradual degradation of performance with increasing  $\Delta t$ . (The one exception to this is the timestep sensitivity seen in DYCOMS; however, my best guess is that this is associated with numerical mixing via advection rather than a decline in stability.) To a good first approximation the cost scales like the number of steps and hence like  $1/\Delta t$ . This scaling is modified slightly because the multigrid solver parameters are automatically adjusted based on the acoustic Courant number.

The 6 s figure for DYCOMS was intended just as an example. However, to avoid distraction, I have replaced that with a statement that all cases become unstable if the timestep is too large. Section 5.4 has been extended to include some discussion of how the cost scales with timestep.

**Semi-Lagrangian accuracy**

• The accuracy of semi-Lagrangian trajectory calculations decreases as eigenvalues of  $\Delta t \nabla \mathbf{u}$  approach/exceed 1.

- Could the author quantifies the impact of these inaccuracies on key diagnostics, such as turbulent kinetic energy or scalar variance, particularly in regions where vertical shear is strongest?

This is an excellent question, related to the discussion under 'Timestep limitations', and certainly deserves some careful study. For now, I can say that often one sees very little dependence of most diagnostics on timestep (again with the exception of DYCOMS) right up to the point where the model becomes unstable; I suspect that this insensitivity occurs because any instability tends to be very localized while the rest of the simulated domain is well behaved.

Comparisons with explicit solvers

• The semi-Lagrangian formulation is noted for enabling large timesteps compared to explicit schemes.

- Could the author provide direct comparisons (e.g., runtime, accuracy, or computational cost) between their method and traditional LES solvers under similar setups?

- How does the semi-implicit scheme affect the resolution of fast dynamical processes, such as wave breaking or sharp density gradients, compared to explicit solvers?

Direct comparisons of runtime and cost are difficult, as noted above in response to the 'General comparisons' question. The discussion in section 5 shows that the results from PTerodaC3TILES are comparable to those of established LES codes, most of which are explicit. The most conspicuous limitations of PTerodaC3TILES are related to its ILES formulation, particularly near the surface. My student Yuhang Tong is making good progress on understanding these limitations and developing mitigations.

The semi-implicit treatment of acoustic waves is not thought to adversely affect results; most LES codes are anelastic or Boussinesq, eliminating acoustic waves altogether. In practice the gravity wave Courant number is generally < 1 except in strongly stable layers at the surface; thus, any slowing of gravity waves by the semi-implicit treatment should be negligible. More likely to be an issue is when semi-Lagrangian advection permits taking timesteps that are long compared to the Lagrangian dynamical timescale, which could lead to under-resolution in time of some dynamical process. However, this scenario is precisely when the deformation Courant number is large, so instability is likely to be as much an issue as loss of accuracy.

Deformational Courant number

• The deformational Courant number is mentioned as a potential limiting factor for stability, especially in regions with strong vertical shear.

- How frequently does the model approach the instability threshold in practical simulations, such as those with strong surface shear or boundary-layer phenomena?

- Are there any mitigation strategies (e.g., adaptive timestep control or trajectory smoothing) to manage cases where the deformational Courant number approaches or exceeds 1?

How frequently the model approaches the instability threshold depends very much on the user's choice of timestep. It is also difficult to monitor because we don't know exactly what that instability threshold is (or what the thresholds are, since there are several potential instability mechanisms).

Some form of mitigation such as adaptive timestep control should be possible, but first we need

a clearer understanding of exactly what limits stability. For example, because of the expense, one would not want to diagnose the eigenvalues of the velocity gradient at every model gridpoint at every step in 'production' runs, but one could imagine doing so to study the model stability. If the stability threshold was found to correspond to the same value of the maximum eigenvalue for different simulations then that would be useful evidence.

I have added a comment to the discussion in section 4.5 to mention the possiblility of some form of mitigation if the causes of instability can be better understood.

**Reviewer 2**

**SUMMARY**

This ms describes a novel and viable LES model including an implementation of consistent moist thermodynamics. It serves as a demonstrator of this thermodynamic formulation in realistic modelling scenarios, represented by a suite of well-known test cases. The formulation is an interesting contribution to model development in this area, and the ms makes a good case for its wider consideration.

The model formulation is presented in sufficient detail (in a section extending to twenty pages) and modelling limitations or areas for future work are well highlighted in general. In the discussion of test cases, the model appears to perform well on the whole. Modelling difficulty in one case, due to the lack of a subgrid turbulence/mixing model, also indicates where future work may be required.

The model is available via the link supplied, along with some further documentation.

Thank you for the positive comments, and for checking the link.

**SPECIFIC COMMENTS**

**1.Introduction**

Lines 60-82: The test advances arguments for the use of a different solution method compared to many "traditional" (line 68) LES codes, and also for numerical diffusion as a substitute for a subgrid scheme, as has been done previously. Is the implication that any diffusive, numerical solution of the unsteady Euler equations constitutes LES?

**I certainly did not mean to imply that!**

(i) A leading order requirement for LES is that there be enough small-scale dissipation to soak up the downscale cascade of enstrophy (2D) or energy (3D). Often that is sufficient to get a superficially plausible simulation, consistent with the idea of an inertial subrange that is independent of the details of forcing and dissipation. However, many other aspects matter for a 'good' LES. (ii) How localised in wavenumber space should the dissipation be? For 2D turbulence, calculation of spectral tendencies at full (high) resolution and at truncated resolution implies that dissipation should be very localised at the end of the resolved spectrum (Thuburn, Kent and Wood, 2014). Typical ILES schemes and parameterisation schemes spread the dissipation over a much wider wavenumber range. I have not seen the analogous results for 3D turbulence, but would be very interested to know if they have been published.

(iii) Backscatter. For 2D, the same spectral tendency calculations show that energy should be scattered back into the energetically dominant large scales. (Again, I would be very interested to see the analogous 3D result.) I have not seen any numerical method that correctly accomplishes the 2D backscatter. The Anticipated Potential Vorticiy Method (Sadourny and Basdevant 1985) dissipates (potential) enstrophy while conserving energy, but backscatters the energy to the wrong scales. (One can, though, construct a backscatter parameterisation that restores dissipated energy to the correct scales.)

(iv) There are arguments that a subgrid model should ideally respond 'dynamically' to the flow and that it should be nonlinear. Advocates of ILES point out that dissipation from the numerical methods does respond dynamically to the flow, becoming zero when the flow is at rest, and can be nonlinear if limiters are used (Monotone Implicit Large Eddy Simulation, or MILES). There is some discussion of these points in the Margolin, Rider and Grinstein (2006) book, but I believe there is scope for further investigation.

(v) I am not aware of any studies that examine in detail how the magnitude and location of dissipation are related to individual flow structures, either for ILES or traditional subgrid models. I think this would be a very interesting comparison. We do know that a numerical advection scheme tends to diffuse structures along streamlines whereas often what is required is diffusion across streamlines; this is related to known limitations of ILES near boundaries.

(vi) All of the above questions are especially challenging in the grey zone, when major flow structures are marginally resolved and there is little or no inertial subrange. Model behaviour can then be very sensitive to details of dissipation (e.g. Tomassini et al 2023) and therefore, by implication, to details of the numerical methods.

References not already in the manuscript:

Thuburn, J., Kent, J., and Wood, N., 2014: Cascades, Backscatter and Conservation in Numerical Models of Two-Dimensional Turbulence. Quart. J. Roy. Meteorol. Soc., 140, 626-638. DOI: 10.1002/qj.2166

Sadourny R., Basdevant, C., 1985: Parameterization of sub-grid scale barotropic and baroclinic eddies: Anticipated Potential Vorticity Method. J. Atmos. Sci. 42: 1353-1363.

The above discussion is too detailed for a model development paper, but I have clarified (around line 80) that there remain many open questions about the strengths and weaknesses of ILES, and that  $PTerodaC^{3}TILES$  is a tool to help answer those questions.

2.Model formulation

Lines 139-142: Compared to the equations of Bowen and Thuburn (2022a), one development in the present model is the introduction of Coriols forces. This is unproblematic for the terrestrial applications discussed later, but would there be a constraint on resolution at very high rotation rates e.g. for possible extraterrestrial applications?

Along with the other non-advective terms, the Coriolis terms are discretized in a Crank-Nicolson manner along trajectories (equations (22), (23)), so will be stable at solver convergence. However, the Coriolis terms are omitted from the linearisation (equations (34), 35)), as is usual in C-grid models such as ENDGame (Wood et al. 2014), because their inclusion would make it difficult to eliminate velocity increments to obtain a Helmholtz problem. Thus, in principle, a value of  $\Omega\Delta t$  approaching 1 might retard (or prevent) solver convergence enough to lead to instability. For global simulations of Earth this can happen with timesteps of around 2 hrs. For the much smaller timesteps of LES a correspondingly more rapid planetary rotation (daylength a couple of minutes) would be needed, so this is unlikely to be a problem in practice.

**In response to reviewer 1 I have added some discussion to section 4.5 of factors that might limit model stability; this is the appropriate place to mention the Coriolis terms too.**

Line 212: It is stated that the model has no representation of subgrid temperature/humidity variability. Presumably this would degrade results if the model is run at low resolution? In the absence of a subgrid turbulence model, subgrid velocity variations will be similarly unrepresented. There is indication in the work of Brown, Derbyshire and Mason (QJRMS 1994) that gridscale stochasticity may be beneficial in modelling stable layers. Would it be possible to introduce a stochastic element in the present model, either with or without a subgrid model?

I agree, I would expect this to degrade results at low resolution, though I have not yet carefully assessed this. Taking into account the subgrid temperature and humidity variability involves various assumptions and approximations, for example a Gaussian joint distribution and a linearization of the saturation curve (e.g. Sommeria and Deardorff 1977). Careful thought would be needed to devise such a scheme that would remain consistent with the thermodynamic potential approach.

I do not see any difficulty, from a model formulation and coding point of view, in including some stochastic terms. A key ingredient in such a stochastic scheme appears to be an estimate of the local, or at least horizontal mean, kinetic energy dissipation rate. An appropriate place to mention a possible stochastic extension, therefore, is in the penultimate paragraph of section 6, which discusses the challenge of estimating numerical dissipation rates.

Sommeria J. and Deardorff J. W., 1977: Subgrid-scale condensation in models of nonprecipitating clouds, J. Atmos. Sci., 34, 344–355.

A possible stochastic extension is now mentioned in the penultimate paragraph of section 6.

Equations (18-20) are not written in an explicitly diagnostic form, although the text immediately preceding makes it seem as though they should be. Maybe that sentence could be rephrased.

Agreed, this could be expressed better. I have rephrased the text.

Subsection 2.7.1 is titled "Consistent surface fluxes of mass and water", and it is possible to overlook the last sentence therein which says that surface entropy flux is treated likewise. Based on the information in the ms, it is not clear (to me) which boundary conditions the model supports. Does it require continuous specification of surface heat and moisture fluxes. Would it also support a constant-temperature boundary? I think this aspect should be made clearer. In Section 3 (line 637) the text refers only to "forcing terms", which does not help to clarify, although lines 979-980 perhaps indicate that surface fluxes are always required.

It is true that this aspect is not as clear as it should be, and I thank the reviewer for pointing that out.

Given the variety of different surface, large-scale, and radiative forcings specified in the different case definitions, I decided to implement these forcings as separate routines for each case (within the module named after the corresponding case), rather than as a small number of general purpose routines with many convoluted options. The surface fluxes of sensible and latent heat are determined by a simple bulk model with specified  $T_s$  and  $q_{sat}(T_s)$  for ATEX and are specified functions of time for the other cases implemented (zero for the BUBBLE and NEU-TRAL cases). The surface fluxes of momentum are zero for the BUBBLE case, given by a simple bulk model for the BOMEX and DYCOMS cases, and given by Monin-Obukhov similarity theory for the ARM, CBL, and NEUTRAL cases. The ATEX case definition calls for a specified friction velocity  $U_*$ . However, under this specification ILES performs poorly, so a simple bulk model is implemented instead. A warning message to alert the user is printed. The specified  $U_*$  formulation can be restored by commenting and uncommenting a few lines of code.

Thus, most boundary conditions that might be desired for a new test case can be implemented by using existing routines as templates. An exception would be if Monin-Obukov theory were required to compute sensible and latent heat fluxes as well as momentum fluxes. The modified formulation presented in Appendix A4 assumes that the surface buoyancy flux is known. Some theoretical work would be needed to extend that formulation to the case of specified surface temperature rather than specified heat fluxes (the Bull and Derbyshire 1990 report would be a good starting point).

The surface fluxes can be switched off via a namelist switch.

I have added some text to the start of section 2.7 summarising the key points of the above.

**6. Conclusion and discussion**

The concluding discussion is (understandably) mainly concerned with summarising model performance and possible next steps in its development. However, given the development and increasing resolution of NWP models, it is possible that these may be used as LES models in the near future. It would be interesting to have the author's opinion on the feasibility of implementing this thermodynamic approach in another existing model such as an NWP model.

To be clear, the thermodynamic potentials approach and the ILES approach are two independent things; PTerodaC3TILES just happens to use both.

As mentioned in the Introduction (lines 30-35), there are two ways one might try to use thermodynamic potentials to obtain consistent thermodynamics.

(a) Use thermodynamic potentials to derive consistent governing equations for the thermodynamics in more or less traditional form, then build a numerical model using those governing equations. It should be relatively straightforward (though still far from trivial) to retrofit this approach to an existing model, and there is interest from some modelling groups.

(b) Build the numerical model directly in terms of thermodynamic potentials, as in PTerodaC3TILES. This alternative has the advantage of being more general and flexible. However, to apply this to an exisiting model, all thermodynamic relations in the model (in parameterization schemes as well as dynamics) would need to be re-expressed in terms of the potentials, requiring pervasive and technically tricky code changes. Thus, realistically, I think this alternative is only likely to be adopted if built into a new model, bottom-up.

Line 897: Could/should the zenodo link also be given as a citation in the bibliography?

Looking at other GMD manuscripts, this seems to be standard. I have included the link as a

citation, as suggested.

A4.Alternative formulation of Monin-Obukhov theory for surface momentum flux

This appendix lists one problem with stable boundary layers (lines 976-977), namely that the lowest model level may not be low enough to lie within the validity range of MOST. It is not obvious to me how a reformulation/approximation of MOST will overcome this problem, other than perhaps making it simpler to apply MOST outside its range of validity? This should be clarified.

Yes, this was not clear enough; thank you for pointing it out. The formulation in Appendix A4 does nothing to extend the range of validity of MOST. It merely ensures that a solution for the surface momentum flux exists and is unique and is vaguely sensible, even outside MOST's range of validity.

I have added some text (second paragraph of Appendix A4) to clarify this point.

Figure A1 does not seem to be referenced in the text. I assume the curves plotted there were calculated for a specific value of surface buoyancy flux, and this value should be given.

Figure A1 is cited at the end of the first paragraph of Appendix A4 and again just after equation (A26).

The surface buoyancy flux is accounted for in the Obukhov length *L* and hence in the nondimensionalization.  $\hat{U}_*$  is a function of  $\zeta$ , while  $\hat{U}$  is a function of *r* and  $\zeta$ . Once we fix  $r = z_0/z$ , as we do in each panel of the figure,  $\hat{U}_*$  is then a unique function of  $\hat{U}$ , with no other parameters to be specified.

**Other author comments**

I am grateful to Yuhang Tong for pointing out that surface momentum fluxes were incorrectly calculated for some cases in the PTerodaC3TILES version 1.0 code. The error affected only those cases that use Monin-Obukhov theory to compute the surface momentum fluxes (ARM, CBL, and NEUTRAL) and involved a missing factor of the von Karman constant in the computation of  $U_*$ .

A bug-fixed version of the code (version 1.2) has been posted on Zenodo.

In the manscript the bug affected the results for the ARM case. I have rerun the ARM case with the bug fixed and have included updated figures (5, 6, 9, 10, 11) and discussion in the revised manuscript as well as linking to the bug-fixed code. Fortunately the differences are rather minor and do not affect any of the main conclusions. They include

\* differences in detail of the cloud field and time series of liquid water content because of the chaotic nature of the turbulent flow (figures 6, 9, 11);

\* some differences in momentum budget (figure 6);

\* the low level peak in *u*-variance is now more in line with the results in Brown et al. (2002), but the peak in *w*-variance at 3 hrs is now smaller than most ensemble members in Brown et al. (figure 10). The peak TKE (figure 9) is reduced to  $1.74 \text{ m}^2 \text{s}^{-2}$ .

The axis units in figure 7(c) and figure 13(c) were incorrect. These have been corrected.